# Imaging single-cell blood flow in the smallest to largest vessels in the living retina

Aby Joseph[1,2]*, Andres Guevara-Torres[1,2], Jesse Schallek[2,3,4]*

[1]Institute of Optics, University of Rochester, New York, United States; [2]Center for Visual Science, University of Rochester, New York, United States; [3]Flaum Eye Institute, University of Rochester, New York, United States; [4]Department of Neuroscience, University of Rochester, New York, United States

**Abstract** Tissue light scatter limits the visualization of the microvascular network deep inside the living mammal. The transparency of the mammalian eye provides a noninvasive view of the microvessels of the retina, a part of the central nervous system. Despite its clarity, imperfections in the optics of the eye blur microscopic retinal capillaries, and single blood cells flowing within. This limits early evaluation of microvascular diseases that originate in capillaries. To break this barrier, we use 15 kHz adaptive optics imaging to noninvasively measure single-cell blood flow, in one of the most widely used research animals: the C57BL/6J mouse. Measured flow ranged four orders of magnitude (0.0002–1.55 µL min$^{-1}$) across the full spectrum of retinal vessel diameters (3.2–45.8 µm), without requiring surgery or contrast dye. Here, we describe the ultrafast imaging, analysis pipeline and automated measurement of millions of blood cell speeds.
DOI: https://doi.org/10.7554/eLife.45077.001

*For correspondence:
aby.joseph@rochester.edu (AJ);
jschall3@ur.rochester.edu (JS)

## Introduction

Retinal neurons have exceptionally high metabolic activity and demand timely and adequate nutrition and waste removal (*Pournaras et al., 2008*). Serving this need, the retinal circulation supplies the bulk of the inner retina with a stratified network of vessels, while the choroidal circulation supplies the photoreceptors. Anatomically, the inner retina is typically fed by a single central retinal artery while return is mediated by the central retinal vein in the holangiotic retina (*Michaelson, 1954*). However, functional characterization of perfusion has been more difficult, especially in the smaller vessels where imaging single blood cells without using invasive foreign dyes requires exquisite spatiotemporal resolution and contrast. Understanding perfusion kinetics at this scale is critical to understanding the entire connected network, especially since blood flow dynamics are tightly linked to physiologically relevant neural loads (neural-glial-vascular coupling). Moreover, studying blood flow at this scale is important to characterize conditions of health and disease which place important constraints on single file blood flow where nutrient delivery and waste product removal take place. In the retina, as in the brain, there is a known linkage between stoppage of blood perfusion and death of neurons. These are manifest not only by transient blockages as seen in branch retinal vein occlusion and central retinal artery occlusion, but also in more insidious diseases such as diabetic retinopathy (*Pournaras et al., 2008*). Unfortunately, the detection of retinal vascular disease in the clinic often comes too late, when either the patient reports vision loss or when structural changes to the vasculature become severe enough to be visualized by clinical ophthalmoscopes. Making matters worse, many vascular diseases of the eye are believed to manifest first in capillaries, which generally evade detailed study due to their microscopic size, thus making early detection in the clinic difficult. Retinal capillaries with single file flow are small, with lumen diameter

**eLife digest** The magical twinkling of the night sky is actually the result of imperfections in Earth's atmosphere. Turbulence in the air distorts the light as it passes through, causing it to bend, which blurs the image. To get a clear picture of distant objects, astronomers use a technique called 'adaptive optics'. Deformable mirrors, controlled by computers, bend the light back to correct the distortion. Now, biologists are borrowing the same technique to take a closer look at the blood vessels of the brain.

At the moment, functional MRI is one of the most popular imaging techniques for measuring blood flow in the brain. But it can only achieve a resolution of around 1 millimetre, and the tiniest capillaries measure less than a hundredth of a millimetre across. These tiny vessels can be examined surgically, or by injecting dyes into the bloodstream, but these techniques carry a risk. As a result, scientists can only use them in experimental animals.

One solution is to look at the vessel network in the back of the eye. The retina is part of the brain and, because the eye is transparent, its blood supply is much easier to see. But imperfections in the eye bend light and blur images of the cells, just like the Earth's atmosphere blurs images of the stars. This is where adaptive optics comes in. Using this technique, it becomes possible to see single red blood cells. Combining adaptive optics with high-speed video could allow us to track cells through the whole network of blood vessels in the eye. At the moment, blood flow measurement techniques either focus on the very big vessels, or the very small ones; they cannot get a complete picture of the whole interconnected system.

Now, Joseph et al. have combined adaptive optics, ultrafast imaging, and a new algorithm to view the whole blood vessel network in the eye of a mouse. The new method can automatically measure single blood cells, even when they are moving at their highest possible speed. It can capture the full range of retinal vessels, from the smallest capillaries to the largest arteries and veins. And, by tracking millions of cells at a time, it can reveal how the pressure wave of the blood changes with each beat of the heart.

The slightest disruption in blood flow to the brain can cause irreversible damage. So, measuring blood flow is crucial to understanding what happens when things go wrong. This new method is not invasive and uses safe levels of near-infrared light which the retina cannot see, making it safe to use in humans. Joseph et al. have already started their first studies in humans, and the baseline data they have obtained for healthy mice also provides a starting point for comparison with mice with genetic mutations or disease.

DOI: https://doi.org/10.7554/eLife.45077.002

ranging from 3 to 7 μm (*Guevara-Torres et al., 2016*). This poses a challenge for conventional imaging systems, such as scanning laser ophthalmoscopy (SLO) and optical coherence tomography (OCT), which do not correct for the aberrations of the eye, and have an optical resolution of 10–13 μm (*Porter et al., 2006*; *Zhi et al., 2012*). Imaging techniques that can visualize the larger vessels (~10–200 μm) are still faced with the problem that single blood cells (~6–8 μm) are densely packed and exhibit non-parabolic velocity profiles, making an accurate assessment of mean velocity and flow difficult without empirical knowledge of the profile shape. Detecting subtle functional changes to flow becomes exceedingly difficult as the ratio of vessel lumen to RBC diameter decreases, because of the particulate nature of blood flow. Due to the above challenges, a device that can non-invasively study single-cell blood flow dynamics in the full range of retinal vessel sizes in the living body has not yet been demonstrated, in either the mouse or human.

Studies in larger retinal vessels using Doppler strategies have provided one solution to report bulk velocity. These techniques, like bidirectional laser Doppler velocimetry, Fourier domain Doppler optical coherence tomography and others, give objective and label free measurements of blood velocity in large and medium-sized retinal vessels in humans (size definitions in Materials and methods) (*Feke et al., 1989*; *Riva et al., 1985*; *Wang et al., 2007*; *Wang et al., 2008*). Based on certain assumptions, the Doppler shift can indicate the speed of optical scatterers. However, applying the Doppler approach to small vessels is challenging as there are far fewer blood cells at the capillary level, thereby reducing the Doppler signature. Additionally, left uncorrected, the smallest

measurement beam that can be formed is limited by the imperfect optics of the eye (~10–13 μm spot size). This limits the readout of vessels to a comparable size. Such techniques also suffer the consequence of contamination of adjacent vessel signal which may be of opposite direction or speed. Axial contributions from deeper vessels may also contaminate single vessel measurement as retinal vessels above and below the interrogation beam can introduce non-specific signals. Doppler OCT has mitigated this issue to some extent; however, signal-to-noise and lateral resolution still limit the signal readout of absolute velocity or flow to only the large retinal vessels in mice, and large and medium-sized vessels in humans. Towards achieving capillary resolution, OCT angiography (OCTA) has recently revolutionized measures of perfused vs. non-perfused capillaries. However, it does not quantitate blood flow or velocity. Conditions of hyper- and hypo-perfusion are not reported and thus early and subtle functional changes in blood flow may be overlooked and only later structural vessel occlusions visualized in disease.

In recent years, adaptive optics (AO) ophthalmoscopy has enabled reporting of single blood cell speed in capillaries in humans (*Bedggood and Metha, 2012*; *de Castro et al., 2016*; *Gu et al., 2018*; *Martin and Roorda, 2005*; *Martin and Roorda, 2009*; *Tam et al., 2010*; *Tam and Roorda, 2011*; *Tam et al., 2011b*) and mice (*Guevara-Torres et al., 2016*), and medium-sized vessels in humans (*Zhong et al., 2012*; *Zhong et al., 2008*; *Zhong et al., 2011*). This has been possible as AO measures and corrects for high order aberrations of the eye, achieving ~1–2 μm lateral resolution *in vivo* (*Liang et al., 1997*; *Roorda and Duncan, 2015*; *Roorda et al., 2002*). Recent advances (*Chui et al., 2012*; *Guevara-Torres et al., 2015*; *Scoles et al., 2014*) in developing phase contrast approaches has enabled visualization of translucent cell properties, like blood cell rheology (*Guevara-Torres et al., 2016*) and blood vessel wall structure (*Burns et al., 2014*; *Chui et al., 2014*; *Chui et al., 2012*; *Sulai et al., 2014*), without the aid of invasive foreign dyes or particles. Recently, we combined this approach with extremely fast camera speeds to resolve densely packed RBCs in single file flow in capillaries (3.2–6.5 μm size) and reported single-blood-cell flux (*Guevara-Torres et al., 2016*) without using exogenous contrast agents.

While the above studies employing adaptive optics have enabled noninvasive measurement of single-cell velocity, measurement of blood flow in the full range of vessel sizes of the mammalian retinal circulation is yet to be achieved. This has partly been a problem of scale as automation is needed to perform quantitative measurements in larger vessels containing hundreds of thousands of blood cells flowing per second. In this study, we provide such a computational approach, thus improving upon seminal adaptive optics strategies (*Tam et al., 2011b*; *Zhong et al., 2008*) which used manual velocity determinations, which could take hours to days of analysis time by a human operator. Lengthy analysis times also preclude the use of such techniques in a clinical setting. In this study, we use the living mouse to benchmark the automation of blood velocity data. The mouse is the most widely used laboratory animal, yet there is a paucity of studies providing measures of retinal blood flow in the same. This gap need be addressed as the mouse has been and continues to be used to model human retinal physiology, including blood flow. The challenge of imaging mouse retinal blood flow is attributed to the difficulties of imaging its rather small eye, with even the largest vessels being only a quarter the size of the largest human retinal vessel. Furthermore, as we detail later in this paper, there is wide discrepancy in the normative values of retinal blood flow reported in the few mouse studies that exist. Given the importance of the laboratory mouse, with its completely sequenced genome and many models of disease, characterization of normative blood flow in the complete vascular tree of the healthy C57BL/6J mouse will propagate future research in a vast number of mouse models of retinal disease and systemic vascular disease. Development on quantification of mouse blood flow approaches will be directly applicable to clinical approaches using the same instrumentation.

In this study, we innovate imaging strategies as well as deploy custom algorithms to image single blood cells noninvasively in the smallest to largest vessels of the mouse retinal circulation (lumen diameters: 3.2–45.8 μm); without requiring exogenous contrast agents to facilitate measurement. Measured flow rate per vessel ranged four orders of magnitude (0.0002–1.55 μL min$^{-1}$) across the complete retinal vascular tree, from the largest vessel near the optic disk to the smallest capillary. In addition to aiding basic science investigation, the noninvasive approach provides a comprehensive array of bioreporters of microvascular perfusion, including: cell velocity, lumen diameter, flow rate, single-cell flux in capillaries, pulsatile and cross-sectional flow, and indices of intrinsic temporal and spatial modulations in flow. To our knowledge, we provide the first measurement of cross-

sectional blood-cell velocity profile and the first absolute measurement of pulsatile blood-cell velocity in the mouse retina. We also provide the first measures of mean (and pulsatile) velocity and flow rate in medium and small vessels (capillaries) of the mouse retina. We found that there is a heterogenous distribution of velocity, flux and other hemodynamic parameters in vessels of varying generation/size, showing that diameter alone is a poor predictor of microcirculation hemodynamics in the central nervous system. Through the innovations presented in this work, imaging the complete and intact perfusion system of the mammalian retina bridges an important gap in studying perfusion dynamics in deep microvascular beds within the living body. Due to the central nervous system origin of the retina, this method provides a powerful way to noninvasively study microvascular integrity and kinetics in a portion of the living brain.

## Results

### Performance, visualization, confirmation and limits of single blood cell velocity determination

Single blood cells were directly imaged using the confocal mode of AOSLO (*Figure 1*). 2D raster video captured at 25 frames per second revealed movement of a dense array of tightly packed RBCs traveling in vessels without requiring contrast agents (*Video 1*). Sequentially capturing 1D line-scans across the same vessel produced a space-time image of moving blood cells with a line resolution of 15 kHz (*Figure 1D*, *Figure 2*). Single cells produce an increase in backscatter providing positive contrast streaks which represent the displacement over time of each cell. Based on the shallowest angle of beam intersection, the maximum velocity that could be measured using this technique was 1275 mm s$^{-1}$, which far exceeds the possible velocities of blood cells in the retinal circulation. Given the liberal range of angles we searched, the velocity bandwidth of the measurements ranged from 0.03 to 1275 mm s$^{-1}$. The Radon transform was able to measure the slope of blood cell streaks within a dense array of overlapping regions of interest (ROIs) in a space-time image (*Figure 3A*). ROIs with dominant streak orientations (*Figure 3B*) produced strong peaks in the Radon sinugram (*Figure 3C*). The angle corresponding to the maximum Radon standard deviation gave the velocity of the cells in the ROI (*Figure 3D*, *Equation. (1)*). As empirical proof of detection accuracy, the measured slope was superimposed on data from a single ROI (*Figure 3B*). While this step is not required, it provided augmented visual feedback to the user which assists in velocity determination and provides a visual confirmation step. The context of velocity as a color code could be understood from naïve users and experts alike. From our data, we observed velocities of up to ~100 mm s$^{-1}$ which represents the upper limit of blood velocity in the eye corresponding to center-line flow of the largest vessels. The lowest velocity was bounded by the motion correction algorithm which adjusted for small eye motion artifacts. Without motion correction, the lowest velocity observed was 0 mm s$^{-1}$ (corresponding to stuck or stationary blood cells). Quality of data was often improved when static features such as the vessel wall and other retinal tissue were removed through a running background subtraction of 10 ms (*Figure 2*), which improved signal detection of blood cells. Even with this motion registration step, the lower velocity bound was still <0.03 mm s$^{-1}$.

The algorithm performed favourably to report only the velocities inside the vessel lumen. While analysis was enabled on the entire image (including portions inside and outside the vessel), only velocities inside the vessel met the SNR calculation (*Equation 3*). By way of SNR calculation, *Figure 3E* shows strong rejection of data outside of the vessel, whereas data inside the vessel provides a strong velocity signature. Similarly, the velocity color map in *Figure 4* shows a clear boundary condition where blood cell velocity is only found inside the boundaries of the vessel lumen. The SNR threshold that matched human performance was at SNR ~ 2.5, meaning a fairly strict criterion measured only the strongest data. Empirically, these confirmation steps provided visual feedback that the algorithm was performing in accordance with subjective observations which is the current standard for such measurements (*Zhong et al., 2008*). Our custom MATLAB code completed automated analysis of a typical space-time image (*Figure 4*) at a rate of ~23.4 ROIs per second (on a 3.7 GHz Intel i7-4820K processor, 16 GB RAM, 64-bit Windows 7). With sparse spatio-temporal sampling, a typical 1 s long space-time image was analyzed in 1.78 min, while still being able to detect pulsatile flow. With dense sampling (high overlap), like that shown in *Figure 4C* and *Video 2*, analysis was

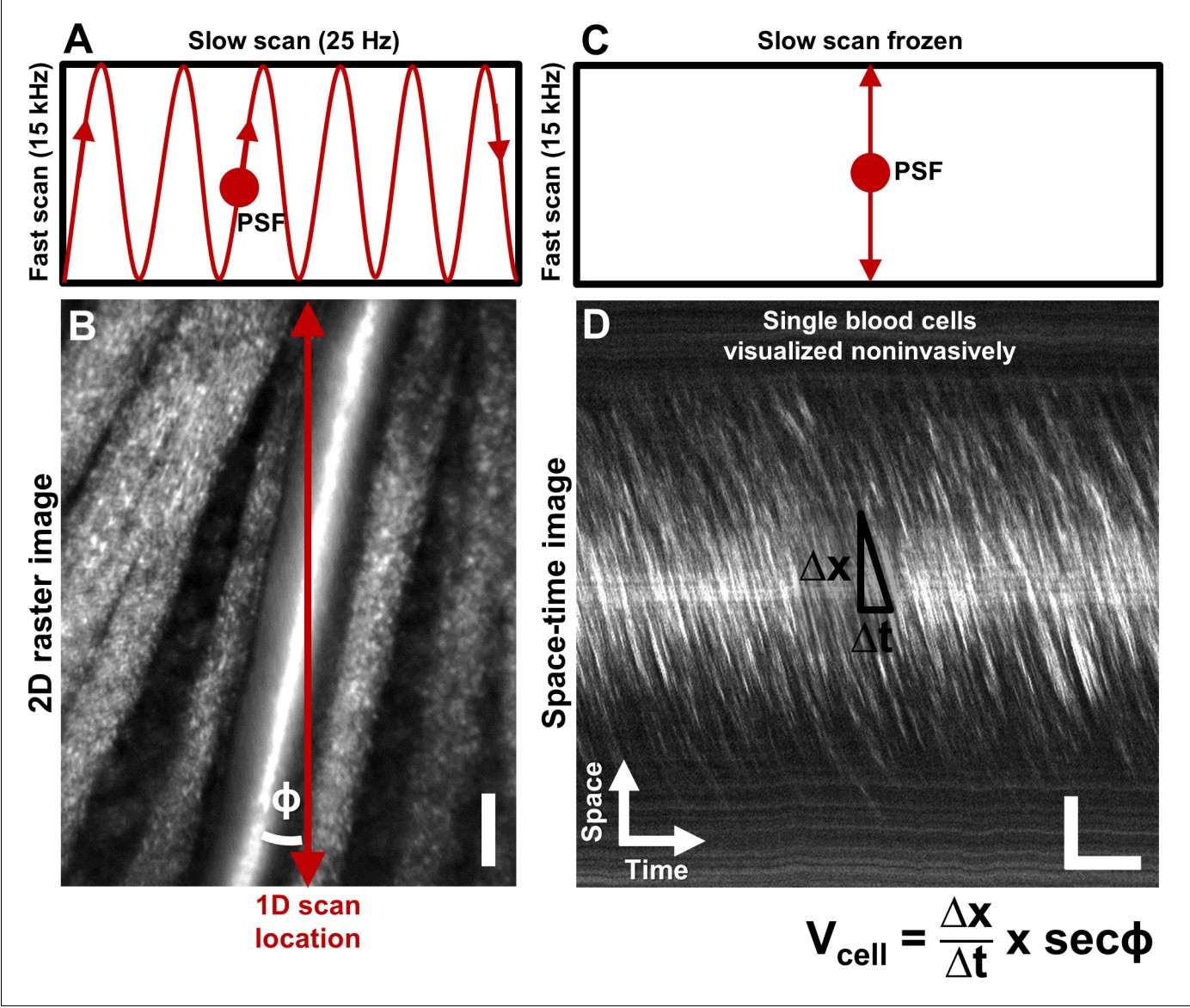

**Figure 1.** Noninvasive imaging of single blood cells and their velocity. (**A**) 2D scan pattern of an adaptive optics scanning light ophthalmoscope (AOSLO) that produces a conventional Cartesian or XY image, shown in B. Two orthogonal scanners raster scan the point spread function (PSF) to produce a video (frame rate = 25 Hz, defined by the slow scan rate). (**B**) A Cartesian image of a first generation arteriole emerging from the optic disk, imaged with 796Δ17 nm direct backscatter. The arteriole (intersected by red arrow) is surrounded by nerve fiber bundles. Typically, 250 frames (10 s) of such a video are recorded (**Video 1**) and averaged to produce the image shown. The scanning field of view was 4.80° x 3.73°. For visualization only, image brightness has been increased by 20–40% in figures of this paper. No brightness or contrast modifications were done for data analysis. Scale bar: vertical = 20 µm (**C**) To directly image all blood cells without aliasing, the scan pattern was modified to freeze the slow scan at a preferred location intersecting the vessel, shown by the red arrow in B. The PSF is now scanned repeatedly in a 1D path at 15 kHz, enabling high temporal resolution and direct quantitative imaging of all biologically possible blood cell speeds in any vessel size in the mouse retina. (**D**) Successive 1D scans are stacked horizontally to produce a space-time or XT image, a small snapshot of which is shown. The white streaks are single blood cells in motion, imaged label–free. The slope of these streaks gives the velocity of the cells along the 1D (fast) scan direction. Correcting for the angle of intersection between the 1D scan and the vessel gives the absolute velocity of the individual blood cells (equation in bottom right is **Equation 1** in text). Visual inspection of the slopes shows that there are fast moving blood cells at the center of the blood column, and slower cells at the edge. 'Stationary' objects, like the nerve fiber bundles, vessel wall or other retinal tissues, manifest as near horizontal features, with near-zero velocity (removed in post-processing by background subtraction, **Figure 2**). Scale bars: horizontal = 5 ms, vertical = 20 µm. Note: In **Figure 1—figure supplement 1**, example of capillary flux imaging is shown, with 1D scan placed orthogonal to the capillary to image individual blood cells and determine exact counts of number of cells passing per unit time.

DOI: https://doi.org/10.7554/eLife.45077.003

*Figure 1 continued on next page*

*Figure 1 continued*

The following figure supplement is available for figure 1:

**Figure supplement 1.** Label-free imaging of single blood cell flux in capillaries.

DOI: https://doi.org/10.7554/eLife.45077.004

completed in 28.25 min for a 1 s long image, with 39,664 ROIs analyzed, providing detailed laminar profile variation with cardiac phase shown in *Video 3*.

## Pulsatile flow measured in all branches of the retinal microcirculation

Data from continuous recordings lasting 1–10 s showed strong modulations corresponding to the cardiac cycle, representing pulsatile flow (shown for a 25.3 µm arteriole in *Figure 4D–F*, *Videos 2* and *3*). The influence of cardiac pulsatility is seen across the vessel lumen, with an average periodicity of the cardiac cycles of 218 beats per minute (bpm) (*Figure 4F*). Pulsatile velocity was observed in all 33 arterioles and venules imaged. Contrary to conventional thought, pulsatile flow was also routinely observed in capillaries (*Guevara-Torres et al., 2016*). Measured heart rate ranged from 186 to 302 bpm across all mice imaged, consistent with the reported range for the anesthetized mouse. To rule out the possibility that heart-rate-induced eye motion contaminated the measures, we tracked the eye motion, in *Figure 5*, from the same vessel shown in *Figure 2* and *Figure 4*. The velocity of the largest eye motion was at least 23 times smaller than the velocity of blood cells (*Figure 5A*). Moreover, the frequency and phase eye motion did not match the pulsatile velocity modulations (*Figure 5B*).

Together, these findings rule out that the velocity signal is an artifact of either respiration or cardiac cycle related eye motion. We confirmed measurements of heart rate with external paw-clip pulse oximetry (PhysioSuite, Kent Scientific Corp., Connecticut, USA) in a subset of vessels. The retinal-vessel measurement of heart rate and an alternate recording performed with the paw-clip pulse oximeter showed strong correlation with correspondence slope of 0.99 (*Figure 5C*, linear fit with forced zero intercept, $R^2 = 0.89$). The lack of perfect fit may arise from running-average of the pulse oximeter measurement, whereas our measurement was nearly instantaneous between adjacent beats.

## Vessel-specific differences in pulsatile flow

While pulsatile flow was observed in retinal blood vessels of all sizes including arterioles, capillaries and venules, there were waveform differences in these branches. Generally, arterioles had a steep leading edge of the velocity waveform whereas venules had a dampened waveform, due to vascular compliance and network hemodynamics upstream and downstream the imaged vessel. Waveforms were analyzed by identifying the peak and trough of the velocity time course corresponding roughly to the systolic and diastolic component of the pressure wave. These control points (*Figure 4F*) served as a phase reference to produce an 'average cardiac cycle' for each vessel, corresponding to 10 s

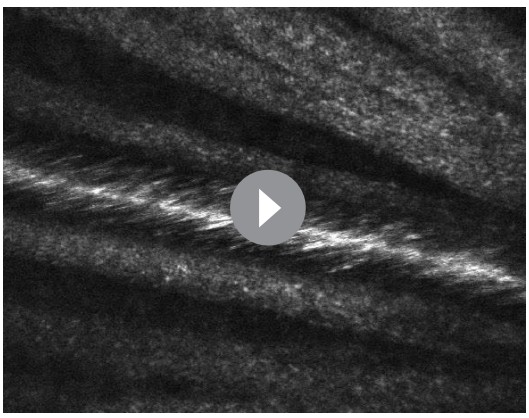

**Video 1.** Video of a first-generation arteriole (25.3 µm) emerging from the optic disk of a mouse, imaged at conventional 25 Hz frame rate using a raster scanning AOSLO. The fast-scan (15 kHz) is in the horizontal direction and slow (25 Hz) in the vertical direction. Direct backscatter is imaged using a 796Δ17 nm super-luminescent diode (~500 µW). Single blood cells can be seen passing as bright streaks within the lumen of the vessel. However, frame-rate of 25 Hz is too slow to quantify the velocity of these fast moving blood cells without aliasing or missing cells. To overcome this problem, 1D scanning is performed at the vessel at 15 kHz to form space-time images of blood cell motion, as shown in *Figure 1* . The video is 10 s long and has dimensions of 4.80˚ x 3.73˚ (163.2 × 126.8 µm). The video has been corrected for eye motion as described in the paper. The 250 frames in this video were averaged to form the high SNR image of the same vessel shown in *Figure 1B*.

DOI: https://doi.org/10.7554/eLife.45077.007

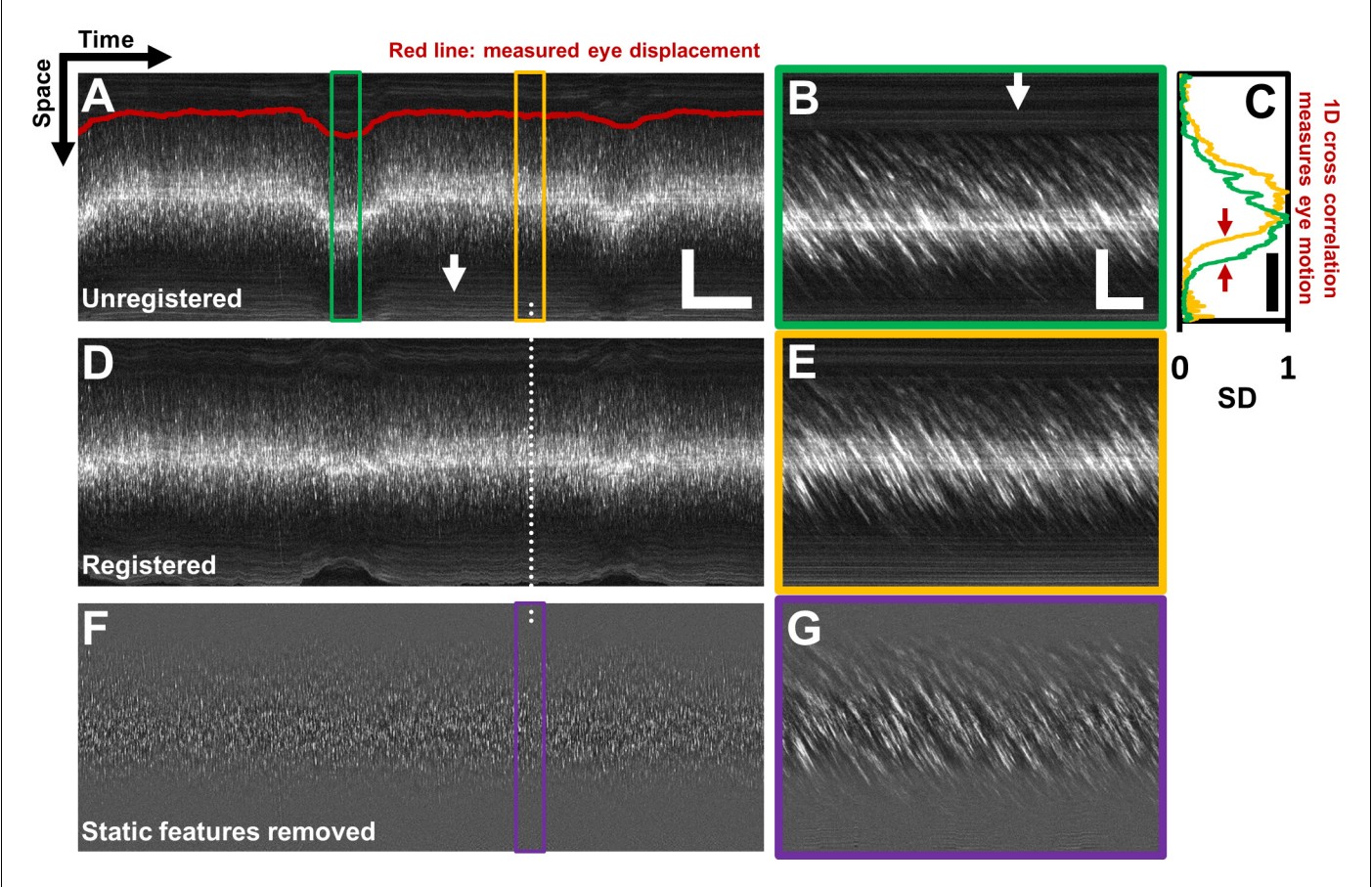

**Figure 2.** Registration of space-time image and removal of 'static' features. (**A**) A one second long raw space-time image from the vessel in *Figure 1*. Actual image is 608 × 15063 pixels across. Eye motion (due to respiration etc.) causes vessel lumen position to change, as observed in image, making laminar profile measurement challenging if eye motion is not corrected for. For visualization, especially of the static features, image brightness of A–E has been increased by 20%. Scale bars: horizontal = 100 ms, vertical = 40 µm (**B**) A sample target image, 40 ms long, which is a zoomed-in version of green box in A. For each line in A, information from a symmetric 40 ms window around it is used as the 'target' for eye-motion registration. Scale bars: horizontal = 5 ms, vertical = 40 µm. (**C**) Standard deviation (SD) of pixel values in time dimension (or 'motion contrast'), is plotted as a function of spatial coordinate. Scale bar: 40 µm. Green and yellow plots correspond to SD profiles of B and E respectively. To register all time points to the same reference lumen position, 1D normalized cross-correlation between SD profiles of target image and a user-defined reference image is used. Thus, eye displacement along fast-scan direction is quantified for each time-point (each line) in space-time image in A. Measured eye motion trace is overlaid on same spatial scale in A (dark red line). This eye motion trace is compared to blood cell velocity trace (measured from same space-time image) later. (**D**) Registered version of space-time image in A. (**E**) A 40 ms long reference image, which is a zoomed-in version of yellow box in A. (**F**) Background subtracted version of image in D. White arrows in A and B shows 'static' features (retinal tissue outside lumen) which move much slower than blood cells do. These static features may interfere with slope measurement of moving blood cells. Therefore, a moving-average window of 10 ms is used to subtract the background, leaving only moving blood cells in the image. (**G**) Background-subtracted version of E. Near horizontal lines due to vessel side-walls, other retinal tissue and specular reflection from top of the vessel have been suppressed.

DOI: https://doi.org/10.7554/eLife.45077.005

of data. The average pulsatile waveform showed a high degree of repeatability. This is shown by way of example in the small error bars in *Figure 4G* which represents n = 35 cycles averaged in a single arteriole. Using vascular indices described by *Equations 4–6* above, we found that in an arteriole and venule of the same vessel generation, the arteriole had a much higher pulsatility index (PI) consistent with physiology of the cardiovascular system (*Figure 4—figure supplement 1*, PI = 0.67 for 25.3 µm arteriole vs PI = 0.18 for 33.5 µm venule). Such functional difference between arterioles and venules can also be visually observed in Figure 9C, where vessels A1 and V1 have similar mean velocity, but markedly different PIs. Furthermore, arterioles routinely showed a temporally asymmetric waveform indicating a bolus pressure wave followed by relaxation of vascular compliance

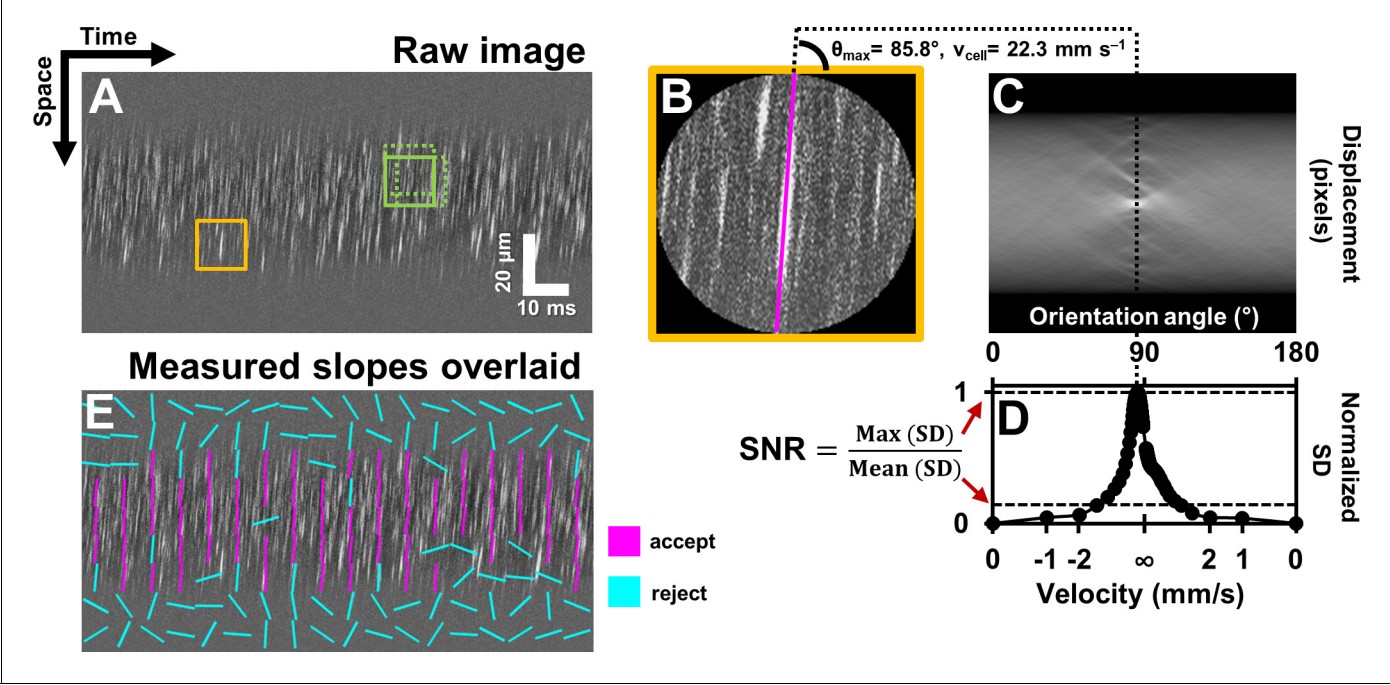

**Figure 3.** Automated measurement of blood velocity using Radon transform. (**A**) A 110 ms long space-time image of a 23.3 µm arteriole scanned obliquely, with static features removed, as described in **Figure 2**. Solid yellow box shows a square ROI of side 157 pixels (~15 µm x 10 ms) used to inspect single-cell velocity. Dashed green boxes show typical 75% overlap of inspection ROIs. (**B**) Zoomed-in version of solid yellow box in A. The ROI is circularly cropped to make the interpretation of the Radon transform easier. (**C**) Radon transform of the ROI in B. Local maxima in pixel intensity correspond to individual blood cell streaks in the single ROI in B. (**D**) Normalized standard deviation (SD) of pixel values of Radon image in C plotted as a function of orientation angle. The horizontal axis is shown both in angle space and corresponding velocity space (angle to velocity mapping given by **Equation 1**). Angle corresponding to peak of variance profile in D gives dominant orientation and velocity of streaks in single ROI in B. Measured cell orientation is overlaid as magenta line in B. The cell was moving at 22.3 mm s$^{-1}$. To determine strength/believability of velocity measurement, a custom signal-to-noise-ratio (SNR) metric is defined as the peak standard deviation (SD) divided by the mean SD. (**E**) Space-time image in A with measured cell orientations in each ROI overlaid using straight lines. For visualization, only a subset of ROIs are displayed, with no overlap. Magenta lines represent ROIs which passed the custom defined SNR threshold (SNR >2.5). Notice that the magenta lines consistently report a tight range of orientation angles, as expected from normal physiology in such a small time epic. Meanwhile, cyan lines often correspond to measured orientations which seem to incorrectly report actual cell orientations, thus showing that the SNR threshold does a good job separating signal from noise. Such dense reporting of single-cell velocity enables measuring subtle fluctuations in velocity due to laminar profile and cardiac pressure wave. Velocities corresponding to measured cell orientations are displayed on a colormap, as shown in **Figure 4**.

DOI: https://doi.org/10.7554/eLife.45077.006

(**Figure 4G**, arteriole asymmetry index (AI) = 2.8). This waveform is contrasted to what is seen in a venule in **Figure 4—figure supplement 1** (venular AI = 1.75).

## Measurement of velocity profile across vessel cross-section

Unlike models of Newtonian/laminar flow, the RBC velocity profile was poorly fit with a parabolic model that is bounded by zero velocity at the lumen surface (**Figure 6**). While often modeled in this way for bulk flow, single RBCs themselves are not expected to be stationary at the vessel wall. A small cell-free zone combined with the vascular glycocalyx represented prevents cell contact with the vascular lumen at the microscopic scale. To quantify the true RBC velocity profile, we fit the data using **Equation 7** (blue line, **Figure 6B**). The $R^2$ of the fit was 0.97, bluntness index B = 1.51 (95% confidence bounds: 1.33, 1.69) and scale factor β = 0.44 (0.38, 0.50). By comparison, a parabolic profile (red line) would have B = 2 and β = 0. An assumption of parabolic flow ($V_{max}/V_{mean}$ = 2) leads to a 39% underestimation in mean flow rate in the vessel shown. While we did not measure plasma velocity, we did model plasma flow between the RBC column and vascular inner wall by assuming linear decline of plasma velocity and enforcing the no-slip boundary condition that may apply to

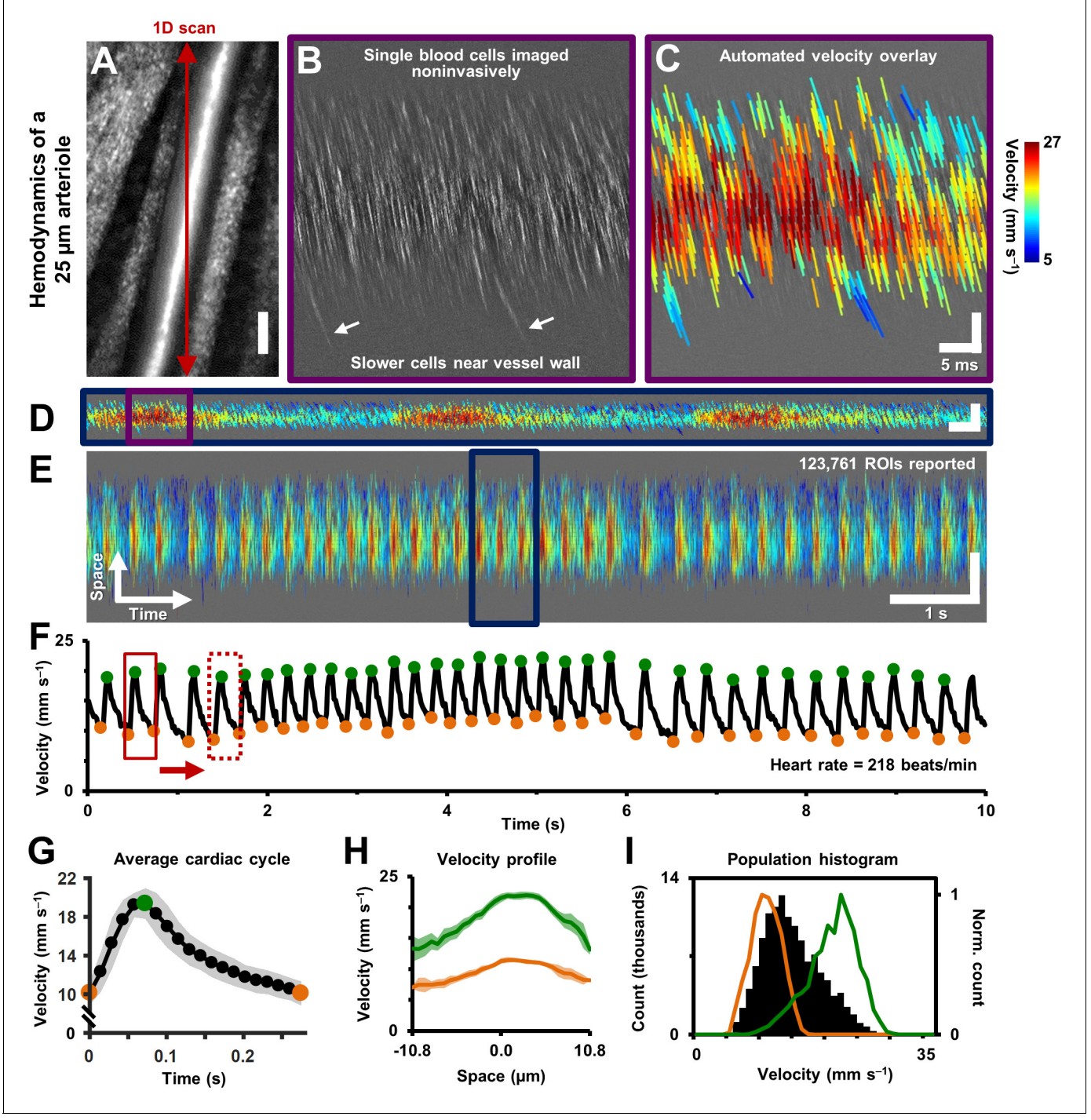

**Figure 4.** Single-cell hemodynamics of a 25.3 µm retinal arteriole, measured *in vivo* and noninvasively. (**A**) Cartesian image of an arteriole. Dark red arrow marks position of fast 1D oblique scanning. Scale bar: 20 µm. (**B**) Space-time image showing single blood cells, imaged with 796 nm direct backscatter. White arrows show slower blood cells near vessel wall. For visualization only, image contrast increased by 40%. (**C**) Automatically measured slopes of cell streaks overlaid on space-time image from B. Color shows absolute cell velocity. Algorithm successfully measured local variations in velocity, including slower cells near vessel wall, marked by white arrows in B. Scale bars: vertical = 20 µm, horizontal = 5 ms. (**D**) Zooming out: space-time image showing ~711 ms of data in the same vessel, with velocity overlaid in color, spanning ~3 cardiac cycles. Time epoch in C is marked with purple box in D. Scale bars: space: 100 µm, time: 25 ms. (**E**) Further zooming out: space-time image showing 10 s of data in same vessel. Time epoch in D is marked with dark blue box in E. Several cardiac pulses can be seen (n = 35). Automated algorithm measured single-cell velocity in 123,761 overlapping analysis regions (ROIs). *Videos 2* and *3* show all ROIs across 10 s with single-cell detail. Scale bars: space: 50.5 µm, time: 1 s. (**F**) Instantaneous velocity vs time: cell velocity data was binned across all space and over a 15 ms time window. Putative systolic (green) and diastolic

*Figure 4 continued on next page*

*Figure 4 continued*

(orange) points shown. (**G**) High SNR 'average' cardiac cycle, which shows highly repetitive pulse waveform, with an asymmetry in time (n = 35 cardiac cycles). Averaging window shown with red boxes in F. Shaded region represents mean ± SD. *Figure 4—figure supplement 1* compares this plot to the average cardiac cycle of a similar sized venule. (**H**) Cross-sectional velocity profile at systolic and diastolic cardiac phases, marked by green and orange time points in G. A blunted parabolic profile is observed, different from prediction of Poiseuille/parabolic flow (vessel lumen diameter = 25.3 µm). Bluntness index of velocity profile changed with cardiac phase ($B_{systolic}$ = 1.67, $B_{diastolic}$ = 1.39). Shaded regions represent mean ± SD. (**I**) Population histogram of all the raw velocity data in E, from 123761 ROIs (in black). Overlaid in color are normalized histogram of cell velocities in all diastolic (orange) and systolic (green) phases.

DOI: https://doi.org/10.7554/eLife.45077.009

The following figure supplement is available for figure 4:

**Figure supplement 1.** Comparison of average cardiac cycle in an arteriole and venule.

DOI: https://doi.org/10.7554/eLife.45077.010

plasma as it contacts the glycocalyx (yellow line, *Figure 6B*). Based on velocity measures above, we observed a gap of 1.8 µm between the position of the most lateral RBCs and the vascular wall.

With dense spatio-temporal sampling, bell-shaped velocity profiles were observed across different cardiac phases (*Figure 4H* and *Video 3*). *Video 3* shows all 20 cardiac phases analyzed. Strongest modulations were observed in the center-line with diminishing modulations closer to the boundary of the vessel. The velocity at center showed the most amount of modulation (*Figure 4H*, 90.5% increase in velocity from diastolic to systolic phase). The velocity modulation at the edge had lower amplitude (71.1% increase in velocity in right lumen edge, from diastolic to systolic phase). Using *Equation 7*, the profile bluntness was observed to change across the cardiac cycle from systolic to diastolic ($B_{systolic}$ = 1.67, $B_{diastolic}$ = 1.39). Combined, the AOSLO measure of blood velocity provides fewer assumptions and populates more free parameters to more accurately model flow in the lumen as a function of space and time.

## Label-free microscopic imaging of lumen diameter

Vessel lumen and vessel walls were clearly visualized using label-free split-detection (SD) imaging (*Figure 7A&D*). The motion contrast images, produced by computing pixel grayscale variance over time, revealed regions of moving blood cells within the vessel (*Figure 7B&E*). The split detection intensity image of the vessel shown revealed a lumen width of 24.9 µm (*Figure 7G*). The motion contrast generated by RBCs revealed a lumen width of 24.8 µm (*Figure 7H*). The simultaneously acquired fluorescein image confirmed a similar lumen diameter of 25.6 µm (*Figure 7I*). The multimodal images, captured simultaneously, show only a small difference in diameter measurement of +3.1%. These small discrepancies are likely real as there is an expected larger diameter measured from plasma fluorescence, spanning the entire lumen, whereas RBCs are absent near the vessel wall due to the known cell-free zone and glycocalyx. Moreover, differences at this scale are small compared to the optical resolution limit of the AOSLO system (~1 µm lateral resolution). Simultaneous label-free split detection and fluorescein imaging confirm that the label-free approach is a valid marker for vessel lumen width, which is important for clinical imaging as it does not require injected dye. Across all 123 vessels imaged, measured lumen diameter ranged from 3.2 to 45.8 µm, representing the entire range of vessel sizes in the mouse retina. For the 33 arterioles and venules, lumen diameter ranged 8.6–45.8 µm.

## Flow in the largest vessels radiating from the optic disk

By combining velocity and vessel diameter data above, flow rate was computed. In primary vessels emerging from the optic disk, within

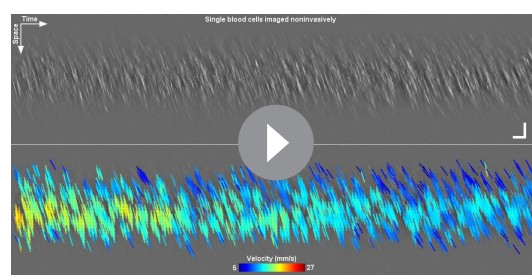

**Video 2.** All 10 s of high-resolution data of single-cell blood flow captured in the 25.3 µm arteriole shown in *Figure 4*. Top: Raw space-time image. Bottom: Cell slopes and velocity overlaid on the original space-time image. N = 35 unique cardiac cycles shown. Scale bars: 4 ms (horizontal), 20 µm (vertical).

DOI: https://doi.org/10.7554/eLife.45077.008

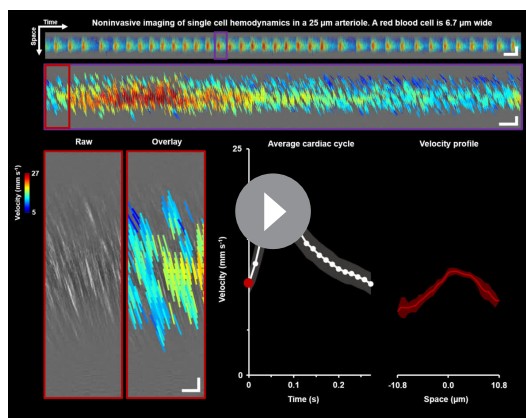

**Video 3.** Blood velocity profile across 20 cardiac phases imaged noninvasively in a 25.3 µm arteriole. Top: Space-time image (10 s) of the 25.3 µm arteriole shown in *Figure 4*. Image is overlaid with measured absolute single-cell velocity, as described in the paper. Scale bars: 300 ms (horizontal), 60 µm (vertical). Middle: Zoom-in of purple box, showing a single representative cardiac pulse in the space time image. Scale bars: 10 ms (horizontal), 25 µm (vertical). Bottom-left: A ~ 12 ms window of the space-time image (red box) is shown in both raw form (showing single blood cell streaks) and in velocity overlaid form. Scale bars: 3 ms (horizontal), 10 µm (vertical). Bottom right: The first plot shows an average cardiac cycle computed from n = 35 cycles, same as *Figure 4G*. The second plot shows the velocity (spatial) profile as a function of cardiac phase. *Figure 4H* had only shown the diastolic and systolic phases. The bluntness and height of the profile can be observed to change with cardiac phase. Bluntness index (B) and relative height of profile edges (β) were quantified using *Equation 7* in the paper. $B_{systolic} = 1.67$, $B_{diastolic} = 1.39$, $\beta_{systolic} = 0.34$, $\beta_{diastolic} = 0.46$. Shaded regions in both plots show mean ± SD.
DOI: https://doi.org/10.7554/eLife.45077.011

a ~170–300 µm radius from the center of the optic disk, the mean flow rate ranged from 0.22 to 1.55 µL min⁻¹ in arterioles (mean (SD) = 0.52 (0.36) across n = 11 vessels), and 0.24 to 1.30 µL min⁻¹ in venules (mean (SD) = 0.57 (0.42) across n = 7 vessels). In *Figure 8B*, mean flow rate, in one mouse, is shown overlaid on structural AOSLO images of these vessels around the optic disk. Assuming 3–7 primary arterioles and venules per mouse retina, we estimated the range for total blood flow to the normal mouse retina to lie between 1.56–3.64 µL min⁻¹ (arteriolar system) and 1.71–3.99 µL min⁻¹ (venular system). This matched range indicates conservation of flow of the source and return flow, further corroborating the precision of the measurement.

## Change in hemodynamics with progression along vascular tree

Several hemodynamic biomarkers were observed to change with vessel generation. In Figure 9A, a montage of structural AOSLO images is shown, with a first-generation arteriole branching off to lower order arterioles, and a first-generation venule, adjacent to the arteriole branch, flowing back into the optic disk. Measured instantaneous velocity and flow rates as a function of time show distinct characteristics of these vessels (*Figure 9C&D*). The mean flow rate decreased with increasing vessel order. Despite being twice as wide as A1, V1 has very similar mean velocity compared to A1, and a much lower pulsatility index. This shows vascular signatures that would be missed by studying only mean velocity or flow, and without temporally gating the measurement with the cardiac cycle. In a different comparison, A1 and A2 have very similar lumen diameters, in apparent contradiction to that predicted by Murray's Law for diameters of parent-daughter vessels. However, a closer inspection shows that flow conservation is not violated: A2 has a lower mean velocity and therefore lower mean flow, necessitated by the two small vessels branching out between A1 and A2. In summary, a functional map across five vessels generations in the same mouse is shown, showing temporal dynamics of microvascular blood flow.

## Conservation of flow and measurement precision

At a branch point (*Figure 9A*), sum of the measured mean flow rates in the two daughter vessels was found to be 0.098 µL min⁻¹, while the measured mean flow rate in the parent vessel was 0.090 µL min⁻¹ (*Figure 9B*). Like the measures of total blood flow in the retina in venular-arterial system, the <8.9% error in these independent measures indicates the precision of the measurement technique. The error may be due to sequential measurements in these vessels, imaged seconds to minutes apart.

## Velocity and flow rate in complete range of retinal vessel sizes

Measured mean flow rate in the complete range of vessel sizes in the retina is shown in *Figure 10B* (123 vessels: 25 arterioles, 8 venules and 90 capillaries across 19 mice. One second of data analyzed

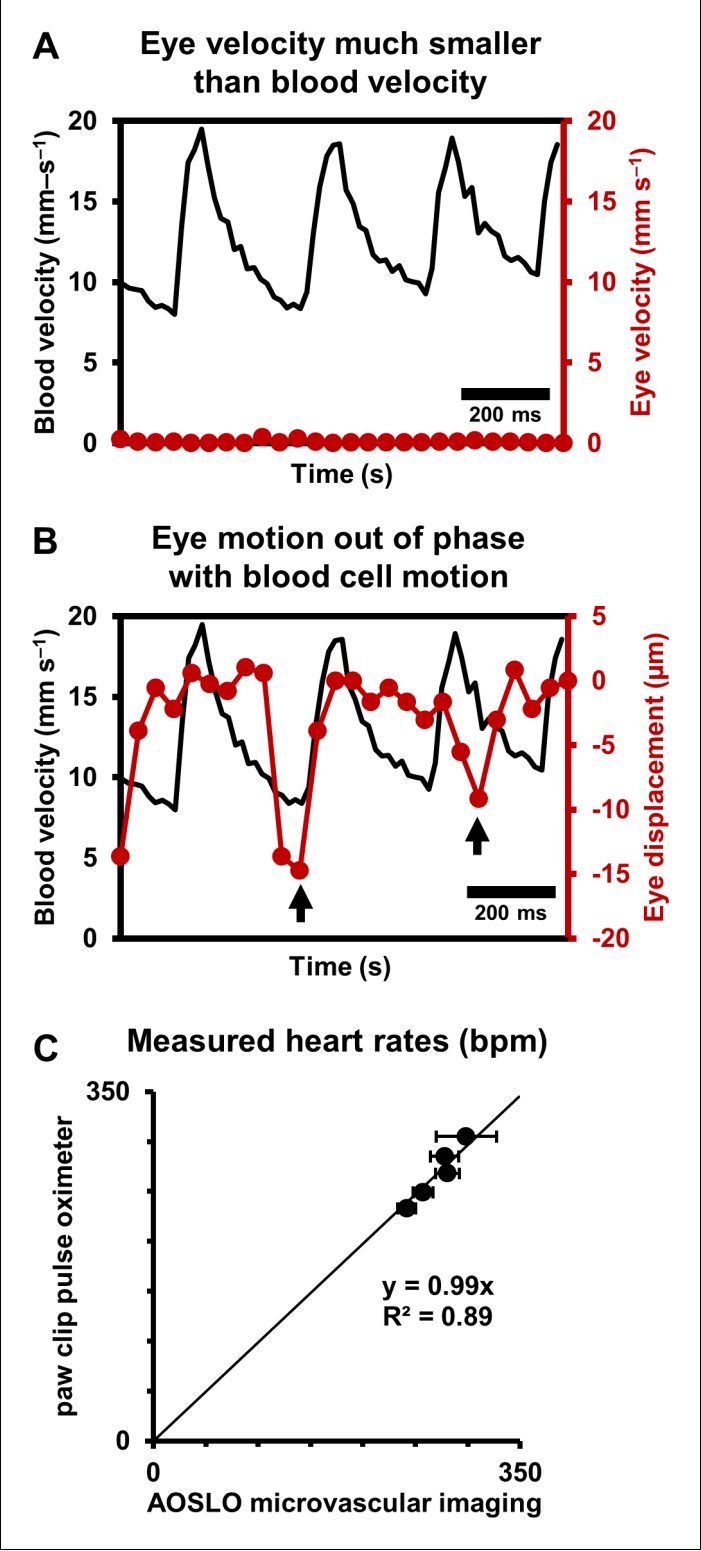

**Figure 5.** Eye motion minimally affects pulsatile blood velocity measurements. (**A**) Comparison of simultaneously measured eye velocity and pulsatile blood velocity (both along fast scan direction). Maximum eye velocity was 23 times smaller than minimum velocity of blood cells, showing that error in blood velocity measurement is less than 4.3%. Data from same vessel as shown in *Figure 2* and *Figure 4* (**B**) Comparison of eye displacement with pulsatile blood velocity. Black arrows point towards two time points when the phase of blood cell motion and eye motion are markedly different, showing that the measured periodicity in blood cell motion is not due to any type

*Figure 5 continued on next page*

Figure 5 continued
of periodic eye motion. (C) Comparison of heart rates measured by AOSLO retinal microvascular imaging and simultaneous paw-clip pulse oximeter (Physiosuite Kent Scientific) measurements. Each data point is a unique vessel imaged for 1 s. Error bars represent standard deviation in instantaneous heart rate (AOSLO) measured in a pulse-by-pulse basis (n = 2 to 4 pulses).

DOI: https://doi.org/10.7554/eLife.45077.012

for each vessel). A large range of mean velocities (0.5–23.1 mm s$^{-1}$) and mean flow rates were measured (0.0002–1.55 µL min$^{-1}$) across the smallest to largest retinal vessels in the mouse retina, thus giving a functional map of the complete vascular network of an intact organ in its native environment. Importantly, mean velocity was observed to be heterogenous across vessel diameters (*Figure 10A* for arbitrary linear fits, arterioles: $R^2$ = 0.31, venules: $R^2$ = 0.43, capillaries: $R^2$ = 0.21).

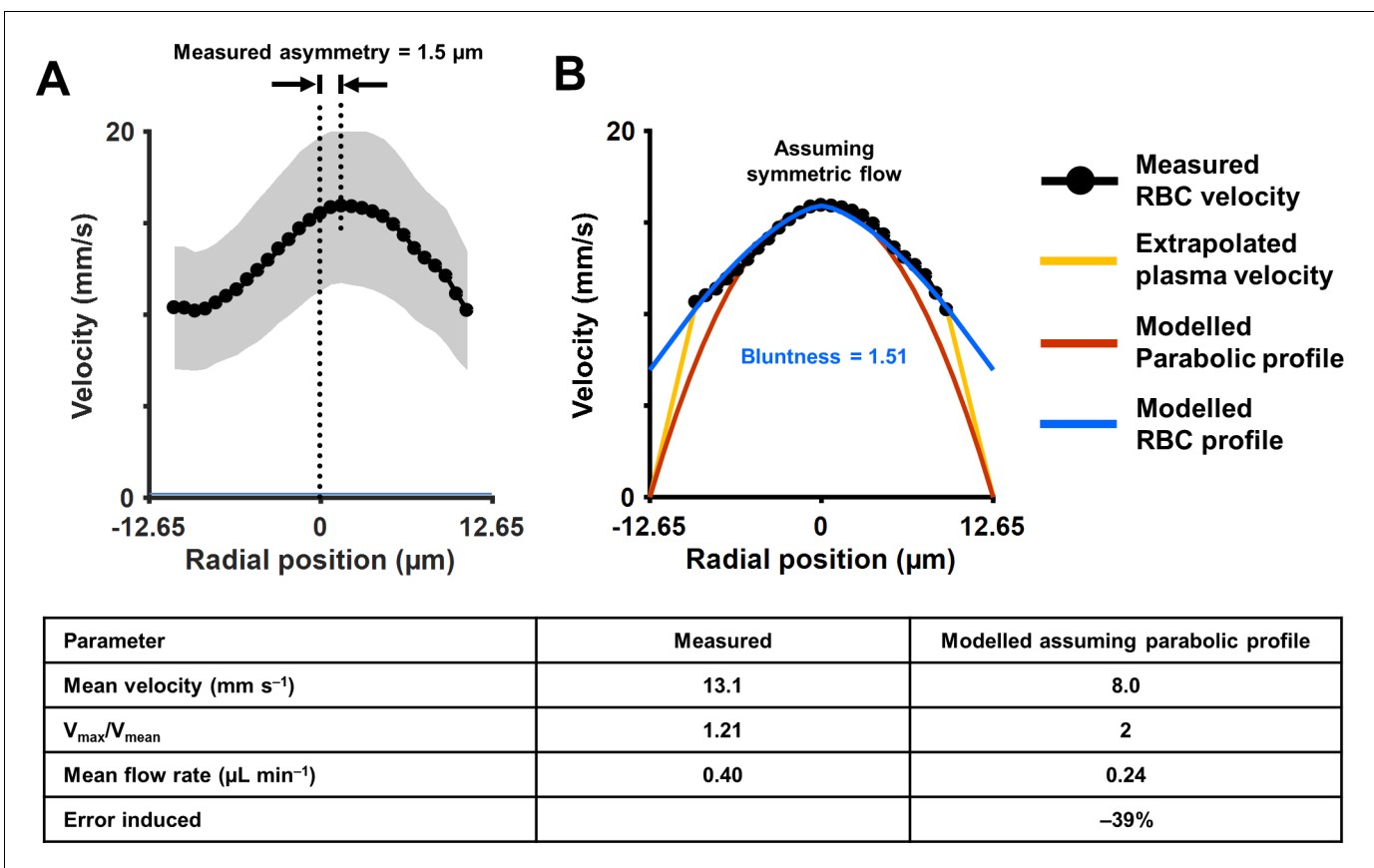

| Parameter | Measured | Modelled assuming parabolic profile |
|---|---|---|
| Mean velocity (mm s$^{-1}$) | 13.1 | 8.0 |
| $V_{max}/V_{mean}$ | 1.21 | 2 |
| Mean flow rate (µL min$^{-1}$) | 0.40 | 0.24 |
| Error induced | | −39% |

**Figure 6.** Measurement of cross-sectional velocity profile of a vessel only a few times larger than an RBC. (A) Measured flow profile, averaged over all cardiac phases in 10 s of data, from the arteriole in *Figure 4* (lumen diameter = 25.3 µm, typical RBC size = 6.7 µm). Profile is a flattened bell-shaped curve, in contrast to predictions of parabolic/laminar profile from Newtonian law. Slight asymmetry is observed (peak velocity is 1.5 µm away from the assigned zero position of vessel, based on the farthest cell velocities that could be reliably measured). A small gap (1.8 µm) between the known lumen diameter and the RBC column is observed, predicted to arise from the known cell-free regions near the vascular wall. Shaded region represents mean ± SD, where variation is across all ROIs in a 10 s window. Number of ROIs analyzed varied from 7333 in the lumen center to 217 near the vessel wall. (B) Assuming symmetric flow, models are fit to the measured flow profile of this vessel. The flatness of the curve is quantified using *Equation 7* (blue line, *Zhong et al., 2011*) (bluntness index B = 1.51, scale factor β = 0.44). A parabolic profile (red line) would have B = 2 and β = 0. The plasma velocity is extrapolated using a linear-decline model (yellow line), to satisfy the no-slip boundary condition at the vascular wall. The bottom table shows that calculation of mean flow rate in the vessel using only centerline velocity and an assumption of parabolic flow profile results in a 39% underestimation of flow, underlining the importance of direct measurement of velocity profiles in vessels this small, and not relying only on models of flow.

DOI: https://doi.org/10.7554/eLife.45077.013

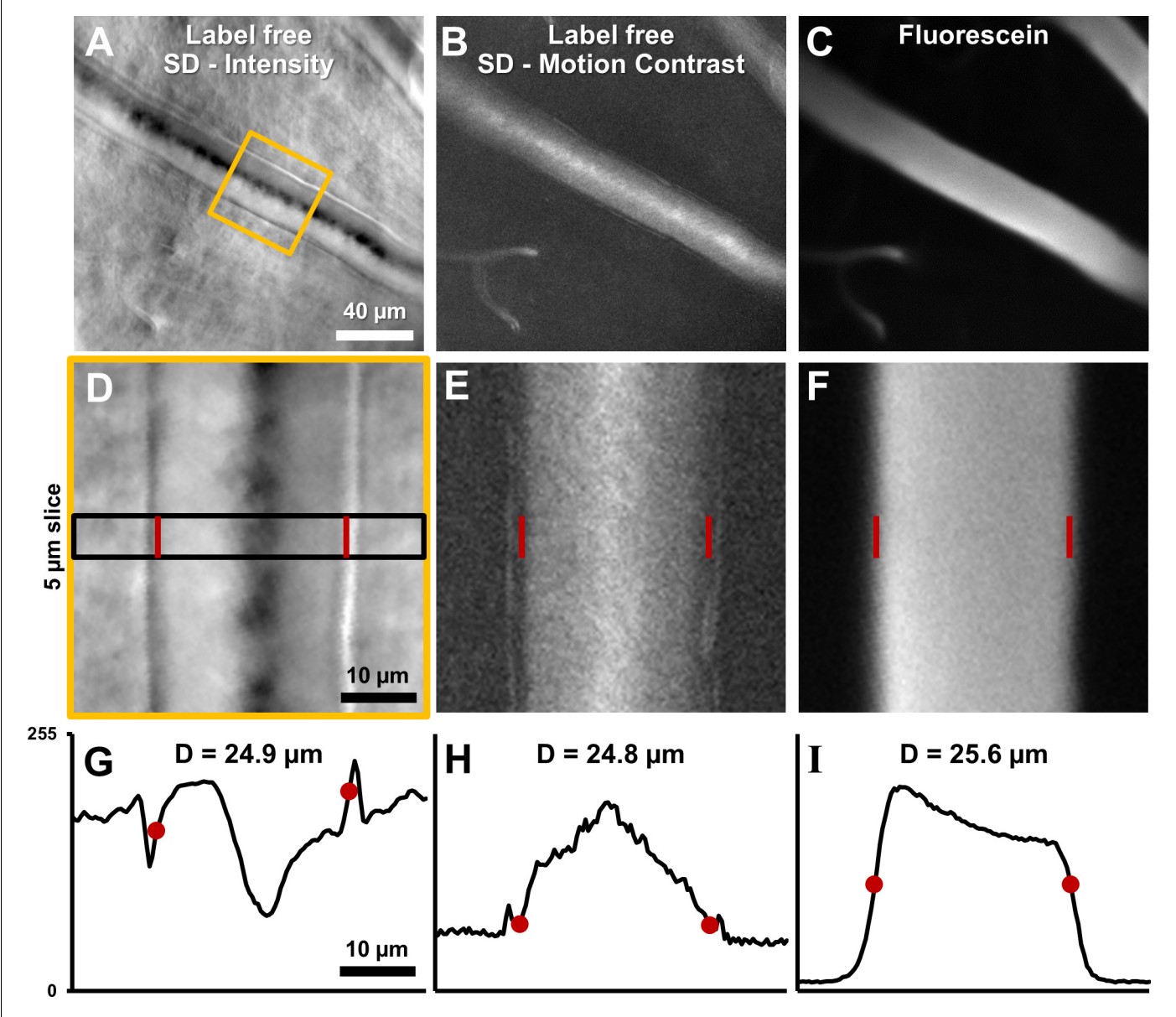

**Figure 7.** Label-free lumen diameter measurement of microvessel matches conventional fluorescein measurement of the same (both modalities simultaneously collected). (**A**) Average intensity image of an arteriole using label-free split detection modality. (**B**) Motion contrast image (temporal standard deviation of pixel values) obtained from eye-motion corrected video from A. Bright pixels show regions of blood cell motion, clearly outlining the lumen of these tiny microvessels. (**C**) Flourescien image of same vessel collected simultaneously. (**D–F**) Zoomed-in images corresponding to yellow box in A, all at the same scale. Radon transform was used to accurately rotate the vessel with its length (and therefore, flow direction) vertical. The vascular wall is clearly seen in D. The side bands in E likely correspond to uncorrected/residual motion of the vascular wall corresponding to respiration/cardiac motion. (**G–I**) 1D intensity profiles in the three modalities computed in a 5 μm slice (shown by black rectangle in D. The lumen boundaries are marked with red circles in G–I and red bars in D–F. Details of these objective measurements are given in manuscript's text. Lumen diameters match across modalities, with errors within 3%. The slightly larger fluorescein diameter is likely attributed to fluorescein representing true lumen width while split-detection motion contrast represents the RBC column width, not including the cell-free zone near vessel walls.
DOI: https://doi.org/10.7554/eLife.45077.014

This shows that diameter alone cannot precisely predict the velocity and related hemodynamic parameters of a vessel, in the entire retinal vascular tree. The calculation of flow requires including the square of the measured diameter (*Equation 9*). Murray's theoretical model for blood vessels predicts a cubic relationship between flow rate and diameter (*Murray, 1926*; *Sherman, 1981*). In close

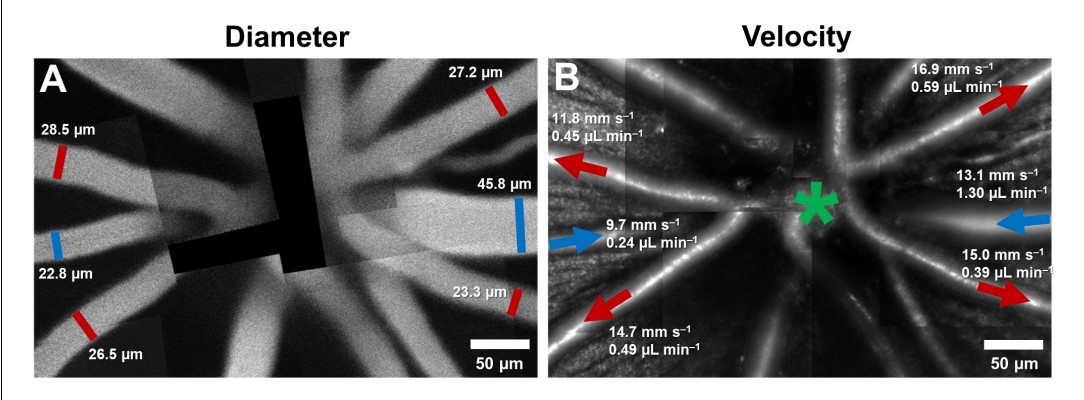

**Figure 8.** Montage of multiple AOSLO fields showing hemodynamics in primary vessels radiating from the optic disk in one mouse. (**A**) Montage of fluorescein images. (**B**) Montage of infrared reflectance images at same retinal location. Green star marks center of optic disk. Lumen diameter and mean flow rate are reported for each analyzed vessel. Arrows point towards direction of flow. Red color represents arterioles and blue, venules.
DOI: https://doi.org/10.7554/eLife.45077.015

agreement, mean flow rate was observed to have a near cubic relation with lumen diameter for medium and large mouse retinal vessels (for power fit, arterioles: exponent = 2.56, $R^2$ = 0.86, venules: exponent = 2.49, $R^2$ = 0.96). Wide heterogeneity was observed in velocity (*Figure 10C*) and flow rate (*Figure 10D*) in capillaries, which have single file flow, supplementing our observation of heterogenous flux across capillary lumen diameters (*Figure 10E*, *Figure 1—figure supplement 1*) (*Guevara-Torres et al., 2016*). (For arbitrary linear fit, velocity vs diameter: $R^2$ = 0.21, flow vs diameter: $R^2$ = 0.61). When plotting cell flux against flow rate for capillaries, (*Figure 10F*, linear fit, $R^2$ = 0.79), the spread in the data represented variations in discharge hematocrit due to plasma skimming. Thus in capillaries, which have single-file-flow, measurement of velocity and flow alone gives an incomplete picture of nutrient delivery; cell flux gives a more accurate representation of the same.

## Discussion

### Summary
Seminal work studying blood flow in the human retina using adaptive optics space-time images (*Tam et al., 2011b*; *Zhong et al., 2008*) used manual determination of cell velocity, which could take hours to days of analysis time due to the problem of scale (hundreds of thousands of blood cell measurements required per vessel for even a few seconds of collected data). Lengthy analysis times preclude the use of such techniques in a clinical setting. In this study, we report single-cell velocity and flow rate in the smallest to largest vessels in the living mouse retina. This was enabled by AOSLO imaging with ultrafast camera capture to noninvasively image blood cells without foreign contrast agents. Pairing this novel imaging paradigm with an automated analysis workflow enabled high-throughput velocity measurement of hundreds of thousands of blood cells per second. Together, these developments provide a new, automated and label-free way of determining vascular perfusion and flow dynamics in the mammal retina, in conditions of health and disease.

For the first time, a functional map of the complete arteriolar tree and venular return of the mouse retina was characterized, with absolute flow measurements performed across all vessel generations and sizes. Pulsatile flow was measured across all retinal vessel sizes, from capillaries to arterioles and venules. This counters the conventional wisdom that capillaries lack pulsatile flow. Quantification of vascular indices like those of pulsatility and profile bluntness across cardiac phases showed utility of measuring subtle temporal and spatial fluctuations of flow, and not just the mean flow, to identify functional signatures that may be disrupted in disease.

The mouse model, with a completely sequenced genome, has become a widely used tool in medical research in the last two decades. Despite this, there is a paucity of retinal blood flow studies in

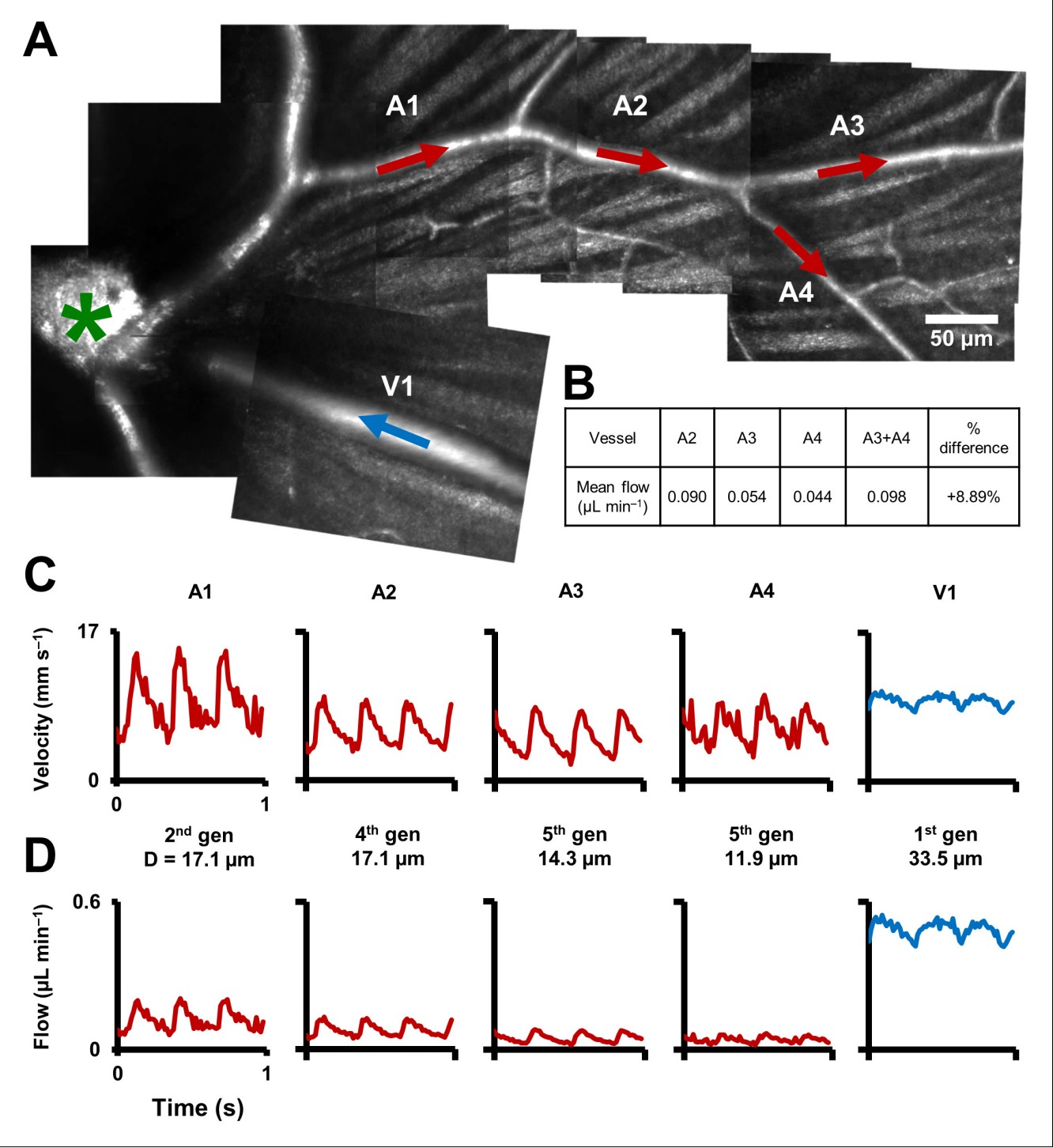

**Figure 9.** Functional mapping of hemodynamics across five vessel generations in the retina. (A) Montage of multiple AOSLO imaging fields showing structure of the mouse retinal vascular tree. The center of the optic disk is marked with a green star. Arterioles in which flow was studied in this mouse are shown with red arrows: A1 (2nd gen), A2 (4th gen.), A3 (5th gen.) and A4 (5th gen.). Blue arrow shows a 1st gen. venule (V1) adjacent to this arteriolar branch. All arrows point toward direction of blood flow. (B) Quantification of conservation of flow at a branch point, in the vessels A2, A3 and A4. Flow is conserved within a < 8.9% error, validating our technique's measurement accuracy in these independent measures made within minutes of each other. (C) Instantaneous velocity vs time plots for the vessels labelled in A. (D) Instantaneous flow rate vs time profiles, derived by multiplying each profile in B with the lumen diameter of the vessel. In summary, a functional map across five vessels generations in the same mouse is shown, showing temporal dynamics of microvascular blood flow.

DOI: https://doi.org/10.7554/eLife.45077.017

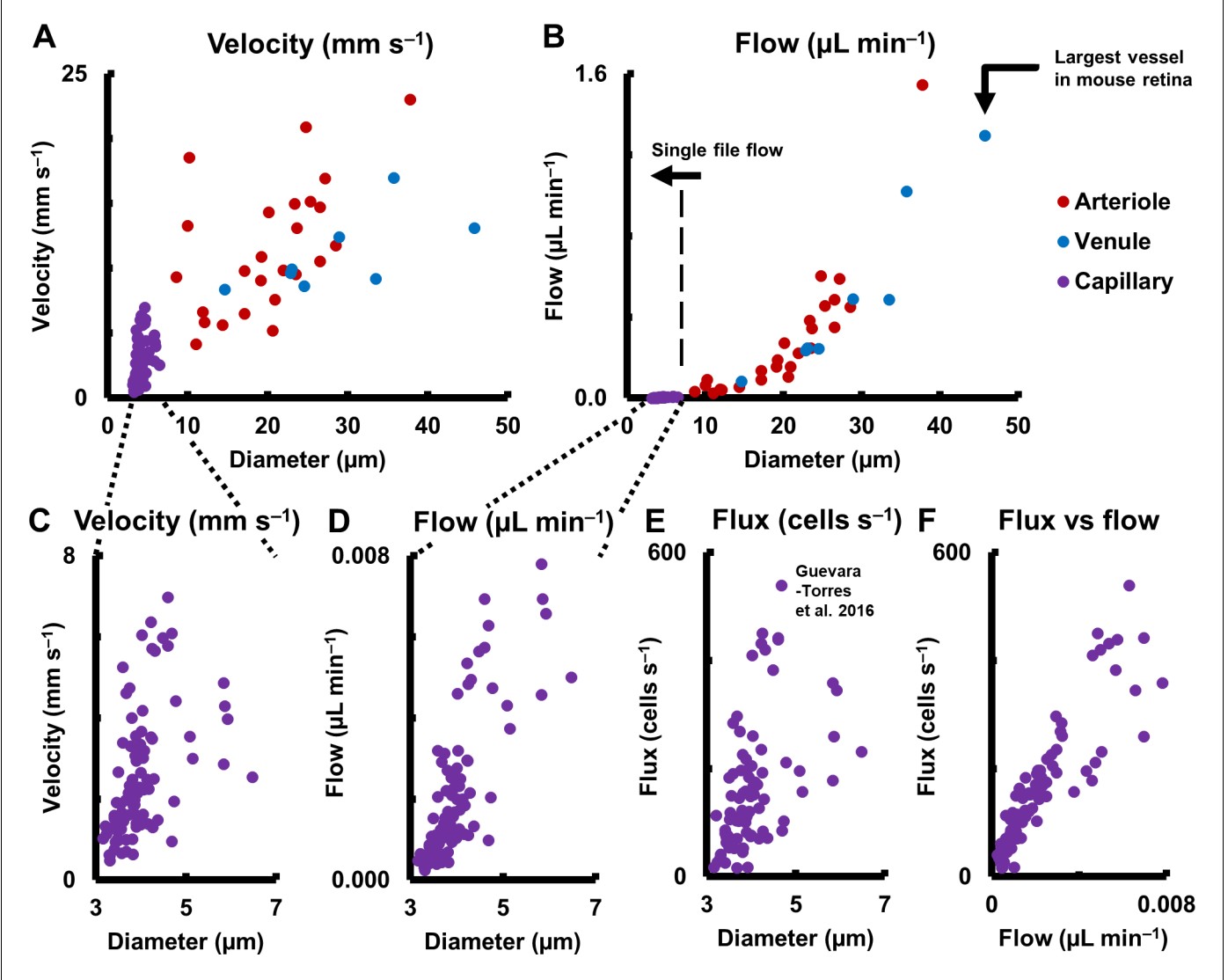

**Figure 10.** Hemodynamics in complete spectrum of retinal vessel sizes in population of vessels (in 19 normal C57BL6/J mice, across 123 vessels: 25 arterioles, 8 venules and 90 capillaries across all vessel generations in retina). (**A**) Mean velocity vs lumen diameter in all vessels. Heterogeneity in velocity is observed across the spectrum (for arbitrary linear fits on velocity-diameter relationship, arteriole: $R^2 = 0.31$, venule: $R^2 = 0.43$, capillary: $R^2 = 0.21$). (**B**) Mean flow rate vs lumen diameter in all retinal vessels, from single-file-flow capillaries to the largest vessel near optic disk in mouse retina. Inclusion of diameter in flow calculation induces strong dependence of flow rate on diameter (for power fit, arterioles: exponent = 2.56, $R^2 = 0.86$, venules: exponent = 2.49, $R^2 = 0.96$). Murray's theoretical model for blood vessels predicts a cubic relationship between flow rate and diameter. (**C–D**) Zoom-in of velocity and flow rate vs diameter for capillaries only (n = 90 capillaries). The linear vertical axis in B prevented the visualization of the near four orders of magnitude range of blood flow rates measured. Wide heterogeneity is observed, with weak correlation of capillary velocity and flow rate with lumen diameter (for arbitrary linear fit, velocity vs diameter: $R^2 = 0.21$, flow vs diameter: $R^2 = 0.61$). (**E**) Data from label-free measurements of single-cell flux in capillaries reported in our previous publication, reproduced here for 90 vessels. (**F**) Correlation of measured flux and flow rate in the same 90 single-file-flow capillaries (linear fit, $R^2 = 0.79$). The spread in the data represents variations in discharge hematocrit due to plasma skimming. Thus, in single-file-flow vessels (i.e. capillaries), measurement of velocity and flow alone gives an incomplete picture of nutrient delivery; cell flux gives a more accurate picture of the same.

DOI: https://doi.org/10.7554/eLife.45077.018

the mouse, as we detail in *Table 1*. Our imaging technique allows recovery of mice after anesthesia, allowing them to be returned to their cages, enabling longitudinal *in vivo* imaging in the same mouse without the need for sacrificing it. Vascular maps obtained during imaging provide the ability to track the same retinal location in the same animal over weeks to months, critical for diagnosing progression and treatment efficacy in mouse models of diseases like diabetes. Moreover,

**Table 1.** Previous studies of vessel diameter, mean velocity and mean flow rate in first-generation retinal blood vessels radiating out of optic disk in normal mice.

Wide disparity in measured values is observed across studies. Figure 11 graphically summarizes differences across studies. Abbreviations: $N_m$: number of mice, $N_v$: number of vessels, D: Mean diameter of vessels, $V_m$: Mean velocity per vessel, $F_m$: Mean flow rate per vessel, TRBF: total retinal blood flow per mouse (calculated by summing flow from all primary arterioles/venules around optic disk), A: primary arterioles, V: primary venules, SD: standard deviation, NOD: Non-obese diabetic. **Special markers:** ˆ Assuming 3–7 primary arterioles and venules each per mouse retina. * SD represents variation across mice and not across vessels. An average arteriolar and venular value was calculated first for each mouse. # values represent mean ± standard error of mean. Variation is across mice and not across vessels. a: Values are mean ± SD, where SD represents variation across vessels, and not mice. Value ranges mentioned in brackets. Measurements in the current study were performed on primary retinal arterioles and venules radiating out of optic disk, at locations within a ~170–300 μm radius from the center of the optic disk. a1: Estimated by multiplying the measured $F_m$ per vessel with the number of vessels per retina. Assumed 3–7 primary arterioles and venules per retina. b: Mean velocities are not reported here as it is unclear if the velocities reported in that study represent the mean or the sum of the individual vessel velocities per retina. c: Variation in $F_m$ per vessel represents variation (SD) across six vessels. Variation in total flow represents variation across three independent measurements. d: Numbers correspond to diameter measurements only. For velocity measurements, $N_m$ = 5–13, $N_v$ = 27–71.

| Study | Technique | Species | Class | $N_m$ | $N_v$ | D (μm) | $V_m$ (mm s$^{-1}$) | $F_m$ per vessel (μL min$^{-1}$) | TRBF (μL min$^{-1}$retina$^{-1}$) |
|---|---|---|---|---|---|---|---|---|---|
| Current study | Adaptive optics line-scan | 15–73 week old C57BL/6J mice | A | 7 | 11 | 25.9 ± 4.6 a (20.1–37.7) | 15.0 ± 4.0 a (9.9–23.1) | 0.52 ± 0.36 a (0.22–1.55) | 1.56–3.64 a1 |
| | | | V | 6 | 7 | 30.6 ± 8.4 a (22.8–45.8) | 11.4 ± 3.0 a (8.6–17.0) | 0.57 ± 0.42 a (0.24–1.30) | 1.71–3.99 a1 |
| *Liu et al., 2017* | Dual ring scanning Doppler vis-OCT | 13 week old non-diabetic TSP1$^{-/-}$ mice | A | 7 | 21–49 ˆ | 40.1 ± 1.8 * | - b | - | 1.86 ± 0.24 * |
| | | | V | 7 | 21–49 ˆ | - | - | - | - |
| *Blair et al., 2016* | Slit lamp biomicroscope | 24 week old C57BL/6J mice | A | 10 | 30–70 ˆ | 26 ± 2 * | - | - | - |
| | | | V | 10 | 30–70 ˆ | 29 ± 3 * | 8 ± 1 * | - | 1.9 ± 0.5 * |
| *Zhi et al., 2014* | En face Doppler OCT/ OMAG | BTBR wild-type mice | A/V | 10 | 30–70 ˆ | - | - | - | 3.05 ± 0.20 # |
| *Wright et al., 2012* | Intravital microscopy | 30–31 week old male C57BL/6J mice | A | 9 | 45 | 57.3 ± 1.1 # | 25.3 ± 1.3 # | 4.02 ± 0.28 # | - |
| | | | V | 9 | 45 | 62.5 ± 2.4 # | 23.2 ± 1.1 # | 4.83 ± 0.41 # | - |
| *Zhi et al., 2012* | En face Doppler OCT/ OMAG | 22 week old female BTBR mice | A | 1 | 6 | - | - | 0.47 ± 0.05 c | 2.82 ± 0.30 c |
| | | | V | 1 | 6 | - | - | 0.55 ± 0.10 c | 3.27 ± 0.28 c |
| *Yadav and Harris, 2011* | Intravital Microscopy | 16 week old male C57BL/6 mice | A | 12 | 36–84 ˆ | 60.0 ± 1.3 # | 30.2 ± 1.4 # | 5.29 ± 0.34 # | - |
| | | | V | 12 | 36–84 ˆ | 67.6 ± 1.6 # | 28.0 ± 1.2 # | 6.39 ± 0.31 # | - |
| *Wang et al., 2011* | Intravital microscopy | 13 week old male C57BL/6 mice | A | 7 | 21–49 ˆ | 56.0 ± 1.1 # | 29.0 ± 0.8 # | - | - |
| | | | V | 7 | 21–49 ˆ | 66.4 ± 2.8 # | 25.1 ± 0.7 # | - | 25.08 ± 1.92 # |
| *Wang et al., 2010* | Intravital microscopy | 13–14 week old male C57BL/6 mice | A | 6 | 30–42 | 54.7 ± 1.0 # | 20.9 ± 0.7 # | 2.98 ± 0.14 # | - |
| | | | V | 6 | 30–42 | 60.8 ± 2.7 # | 21.0 ± 0.7 # | 3.86 ± 0.36 # | - |
| *Wright et al., 2009* | Intravital microscopy | 15–16 week old male C57BL/6 mice | A | 7–8 | 28–49 | 59.4 ± 0.9 # | 28.3 ± 1.4 # | - | 26.34 ± 2.52 # |
| | | | V | 7–8 | 28–49 | 69.5 ± 1.4 # | 26.3 ± 1.2 # | - | 31.80 ± 2.40 # |

*Table 1 continued on next page*

*Table 1 continued*

| Study | Technique | Species | Class | $N_m$ | $N_v$ | D (µm) | $V_m$ (mm s$^{-1}$) | $F_m$ per vessel (µL min$^{-1}$) | TRBF (µL min$^{-1}$retina$^{-1}$) |
|---|---|---|---|---|---|---|---|---|---|
| *Lee and Harris, 2008* | Intravital microscopy | 16–30 week old female euglycemic NOD mice | A | 5 | 9–13 | 61.0 ± 1.5 # | - | 3.36 ± 0.18 # | - |
| | | | V | - | - | - | - | - | - |
| *Wright and Harris, 2008* | Intravital microscopy | 16 week old C57BL/6 mice | A | 7–9 d | 41–50 d | 60.4 ± 1.1 # | 23.0 ± 0.5 # | - | - |
| | | | V | 7–9 d | 37–49 d | 71.4 ± 2.9 # | - | - | - |
| *Ninomiya and Inomata, 2006* | Scanning electron microscopy (*ex vivo*) | 16 week old male mice | A | 10 | 80 | <28 | - | - | - |
| | | | V | - | - | - | - | - | - |

DOI: https://doi.org/10.7554/eLife.45077.016

measurements were performed using safe infrared light levels, to which the eye is insensitive, setting the stage for translation to clinical evaluation. Toward this end, we have shown pilot data of the same technique applied to humans with automated velocity measurements performed in retinal vessels of intermediate size (*Joseph et al., 2018*).

## Velocity measurement algorithm and its key assumptions

The custom Radon transform algorithm enabled precise, automated and objective measurements of absolute blood cell velocity from space-time images, including subtle spatio-temporal modulations in velocity which may be missed by subjective evaluation with the unaided human eye. To augment such modulations, that reveal vascular physiology, the algorithm displayed a colored overlay of measured cell slopes, enabling visual comparison to actual slope orientations, and providing visual feedback on measured velocities using a color scale. The small size and dense sampling of the analysis ROIs ensured single-cell measurement of blood velocity, and its subtle modulations due to cardiac pressure wave and cross-sectional flow profile. The non-linear mapping of search angles and velocity ensured that only physiologically relevant angles were searched by the algorithm, reducing computation time while not sacrificing on measurement precision. Thus, while slow moving blood cells were coarsely sampled, fast moving blood cells were sampled finely enough to not miss subtle fluctuations in velocity. Each analysis ROI contains some axial contributions from slower moving blood cells above and below the interrogation beam, as previous studies in humans using similar adaptive optics scanning techniques have rightly identified (*Zhong et al., 2008*). This was mitigated in our study due to two reasons: First, the ~2x NA of the mouse eye gives a ~4x improvement in axial resolution compared to the human eye (*Geng, 2011*). Second, and more importantly, in a given ROI, the velocity reported by our Radon transform analysis is dominated by features with the highest contrast, i.e. the cells at the focal plane. The focal plane bisected the vessel at its widest point.

## Utility of measuring particulate flow

Why measure blood flow at a single-cell level? Measuring flow in vessels whose size is of the order of the size of a red blood cell necessitates measuring cellular scale or particulate flow. The average undeformed size of an RBC is around 6.7 µm in mice (*Gulliver, 1875*) and around 8.0 µm in humans (*Benga et al., 2000*; *Jay, 1975*), and retinal vessel lumen sizes vary from ~3–46 µm in mice (this study) and ~4–200 µm in humans. In vessels this small, the flow dynamics deviate from simple Newtonian or Poiseuille flow. Our direct cell-by-cell measurement of particulate blood flow opens new possibilities to study flow dynamics in the full range of retinal vessels sizes, enabling detailed study of laminar flow in the larger vessels and the complex nature of non-Newtonian and single blood cell rheology at the level of small capillaries. At the capillary level, this direct measurement is critical as simple diameter-resistance models poorly represent the particulate nature of single-cell blood flow. At this level, the vessels lumen is often the size, if not smaller than the undeformed red blood cell (*Skalak and Branemark, 1969*).

In addition to the bio-rheological shear that is observed at this level, measurement of blood flow in vessels of only slightly larger diameter also necessitates consideration of the biphasic nature of blood flow composed of cell and plasma flow (*Secomb, 2017*). A well-known aspect of such consideration is the aggregation of blood cells in the central portion of the vessel lumen, with a thin cell-free plasma layer next to the vessel walls (*Fåhraeus, 1929*; *Fåhræus and Lindqvist, 1931*; *Hochmuth et al., 1970*; *Sutera and Skalak, 1993*), first reported by Poiseuille in 1835. The width of the cell-free layer can lie anywhere between 0.4 to 2.1 µm in glass capillaries of size 4.5–9.7 µm (*Hochmuth et al., 1970*), depending on blood velocity and not taking into account the 0.4–1.0 µm thickness of the glycocalyx in real capillaries (*Pries et al., 2000*; *Vink and Duling, 1996*). In larger vessels of the microcirculation, the layer can be between 0.9 and 5.4 µm in 10–40 µm vessels, depending on hematocrit (*Fedosov et al., 2010*). The existence of the cell-free layer leads to a non-zero velocity of blood cells closest to the vessel wall, with the plasma velocity presumably sharply decreasing to zero at the vessel wall to satisfy the no-slip boundary condition of fluid flow. One consequence of these effects is a blunted cell velocity profile, considerably deviated from the prediction of a parabolic profile from Poiseuille flow.

To put the above in the context of our measurements because our measurements are focal and represent single-cell velocity across the vessel cross-section, fewer assumptions are needed to accurately model flow. Examples of this are seen in *Figure 6* which removes the assumption of parabolic flow and reports more precise, blunted, plug flow. This is an important distinction of measures of whole blood flow (containing plasma) vs blood cell velocity which is particulate in nature. As shown in *Figure 6B*, an error of –39% would occur if mean flow was incorrectly calculated by measuring only centerline velocity and assuming parabolic flow. Direct measurement of flow profiles in vessels only a few times wider than an RBC also opens the possibility of measuring shear rate, which is proportional to the ratio of mean velocity and diameter, and wall shear-stress (WSS) which is proportional to the slope of the fluid velocity profile at the vessel wall (or in the cell-free layer, if the slope is constant within the layer). WSS is an indicator of vessel wall health and vascular remodeling/angiogenesis. In disease, disruptions in blood flow and WSS lead to microaneurysms in diabetes and plaque formation in atherosclerosis.

We measured particulate flow in the retinal vascular tree in two modalities. In vessels with >7 µm diameter, we measured velocity and flow with oblique scanning. In vessels with single file flow (capillaries with <7 µm diameter), we measured flux, velocity and flow, with orthogonal scanning (*Figure 1—figure supplement 1*). Compared to flow rate (µL min$^{-1}$), flux (cells s$^{-1}$) gives a more complete description of nutrient delivery at the level of capillaries, which have heterogenous local hematocrit.

## Cross-sectional blood-cell velocity profile measured for first time in mouse retina

By measuring particulate flow, we quantified hemodynamic metrics never before measured in the mouse. Mouse retinal vessels are small- the inner diameter of even the largest of them is only about a quarter of that of the largest human retinal vessels. To our knowledge, blood-cell velocity profiles have never been measured in the mouse retina, in any vessel size, because of limitations of imaging resolution. For the same reason, it has always been difficult to quantitatively measure such a profile in small retinal microvessels, in any species. Investigators have used adaptive optics to report velocity profile as a function of cardiac phase in vessels as small as a 72 µm artery in the human retina (*Zhong et al., 2011*). We report the first measurement of velocity profile as a function of cardiac phase in a mammalian retinal vessel as small as 25 µm. This is important since such functional changes become harder to detect as the ratio of vessel lumen to RBC diameter decreases. Furthermore, while the definition of a profile becomes less clear as the vessel diameter approaches that of blood cells, we measured decreased velocity at the edge compared to the vessel center in a vessel as small as 10.2 µm. Measured profiles, even in the largest mouse retinal vessels, were blunted and not parabolic, consistent with a thin cell-depleted layer and the Fåhræus and Fåhræus-Lindqvist effects in the microcirculation.

## Pulsatile blood-cell flow measured for first time in mouse retinal arterioles and venules

Beyond spatial dynamics (velocity profile), we measured temporal dynamics in small vessels of the retinal microcirculation. Pulsatility indices in the microcirculation can be biomarkers of diseases like atherosclerosis, hypertension and diabetes. Multiple investigators have measured pulsatile flow in the human retinal microcirculation using adaptive optics, including pulsatile leukocyte velocity in human capillaries (*Martin and Roorda, 2009*; *Tam et al., 2011a*; *Tam et al., 2011b*), pulsatile erythrocyte velocity in human capillaries (*de Castro et al., 2016*; *Gu et al., 2018*), and pulsatile erythrocyte velocity in medium-sized human arterioles/venules (*Zhong et al., 2012*; *Zhong et al., 2008*; *Zhong et al., 2011*). Our previous report of pulsatile erythrocyte flux in a 3.6 μm mouse capillary (*Guevara-Torres et al., 2016*) was the first measurement of pulsatile cell flow in a mouse retinal vessel. In the current study, we report pulsatile cell velocity in medium and large mouse retinal vessels for the first time, with much reduced pulsatility index measured in venules, thus connecting the dots and demonstrating direct measurement of pulsatile flow in the complete retinal vascular tree. The lack of observed pulsatile flow in venules (*Feke et al., 1989*; *Riva et al., 1985*) and capillaries could be explained by limitations of previous approaches that lacked sensitivity to subtle velocity fluctuations in small vessels. Measurement of pulsatility biomarkers using our approach enables early detection of functional changes in these tiny vessels in microvascular diseases.

## Beyond total retinal blood flow - studying complete range of vessel sizes

In the mouse, the total retinal blood flow (that is, the total supply to the inner retina, obtained by summing the mean flows from all primary arterioles or venules) has been a popular biomarker for evaluating the healthy and diseased retina (*Table 1*). While the total blood supply to the retina is one biomarker of vascular health, changes happening in lower vascular branches in disease may be missed. This particularly becomes a problem when early pathological changes in smaller vessels do not result in changes in bulk flow rate in larger parent vessels, due to either vessel anastomosis or individual changes cancelling out.

In an interconnected vascular network, a disruption in flow at any point can affect other parts of the network. This basic feature of a flow network is at the root of the difficulty in studying many vascular diseases of the eye. Disruptions in flow at different locations on the vascular tree can have a spectrum of severity and impact on the health of the retina. For example, a disruption at the small capillary level can cause a redistribution of local blood flow. At low severity levels, capillaries adjacent to the affected capillary may be able to share some the increased workload. Such local disruptions at the microvascular level have even been reported in the healthy retina with no known disease (*Guevara-Torres et al., 2016*; *Schallek et al., 2013b*). As the severity and frequency of such local disruptions increase in disease, a tipping point may be reached when neighboring capillaries can no longer accommodate the extra workload they receive from decreased or no perfusion in affected capillaries. This can lead to a cascade of events where larger vessels that feed and drain these small capillaries begin to have abnormal perfusion and subsequently, structural changes. Such a sequence of events is one hypothesis of the mechanism of progression of diabetic retinopathy. Vascular diseases of the eye can also begin at the level of large arterioles and venules, like in branch retinal vein occlusion, where watershed areas of a large venule are adversely affected. Disruptions at different hierarchies in the retinal vascular system can also have a large temporal bandwidth. Stopped or reduced perfusion in large vessels can have a relatively quick (seconds to minutes) impact on smaller vessels and tissue in the watershed area of the affected vessel. On the other hand, disrupted perfusion at the level of small capillaries can have a slow but severe influence (hours to years) on the rest of the retinal tissue, as is believed to be the case in diabetic retinopathy.

Our approach addresses some of these challenges by using the same imaging instrument to get measures of absolute single–cell blood flow in the smallest to largest retinal vessels in the living retina. This was made possible by using a high bandwidth measurement technique (measurable particle velocities: 0.03 to 1275 mm s$^{-1}$). Imaging techniques with a temporal resolution of even a few hundred frames per second (*Gu et al., 2018*) are insufficient to track velocity in medium and large retinal vessels. We provide the first normative database of mean velocity and flow rate in vessels other than first generation vessels in the mouse. Investigation of both mean flow rate and spatio-temporal

dynamics in multiple vessels generations (*Figure 9 and 10*) sets the stage for longitudinal study of the progression of 'slow' diseases like diabetes across the complete retinal vascular tree.

## Discrepancy in mouse literature in velocity and diameter measurements in first-generation vessels

The flow through a vessel varies as the square of the inner lumen diameter (*Equation 9*), underlining the need for careful measurement of the vessel lumen diameter to get accurate values of flow rate. While investigating the mouse literature for normative values of diameter, velocity and flow in retinal vessels, we found, first, that there is a paucity of such studies. To the best of our knowledge, we have counted all of 15 journal papers which measure retinal blood velocity and/or flow in the living mouse eye (*Blair et al., 2016*; *Harris et al., 2013*; *Lee and Harris, 2008*; *Liu et al., 2017*; *Muir et al., 2012*; *Wang et al., 2010*; *Wang et al., 2011*; *Watts et al., 2013*; *Wright and Harris, 2008*; *Wright et al., 2009*; *Wright et al., 2012*; *Yadav and Harris, 2011*; *Zawadzki et al., 2015*; *Zhi et al., 2014*; *Zhi et al., 2012*). As stated earlier, all these studies measure velocity/flow only in the primary (first generation) vessels emerging from the optic disk. Studies with objectively report-able values are summarized in *Table 1* (except for *Watts et al., 2013*). A graphical summary, classi-fied by imaging technique, is given in *Figure 11*. In our literature survey, some important points were observed. There is disagreement across previous studies in the values of measured dimeter, velocity and flow of first-generation vessels in the mouse retina. In particular, about half of the stud-ies, which use intravital microscopy, report vessels diameters that are ~2 times greater, mean veloci-ties that are ~2 times greater and mean flow rates that are ~8 times greater than that of all other techniques, including our own (*Figure 11*). Our values of mean diameter of primary vessels matches that of the *in vivo* slit lamp biomicroscope study (*Blair et al., 2016*) and is within the range reported by *ex vivo* scanning electron microscopy (*Ninomiya and Inomata, 2006*). Our values of mean flow rate matches that reported by en face Doppler OCT/OMAG studies (*Zhi et al., 2014*; *Zhi et al., 2012*). Comparing cited values for the total blood flow rate to the retina shows the island of values formed by all studies in the range of 1.56–3.99 µL min⁻¹, in contrast to the range of 25.08–31.80 µL min⁻¹ reported by intravital microscopy (*Figure 11*). Early signs of these inconsistencies in the mouse

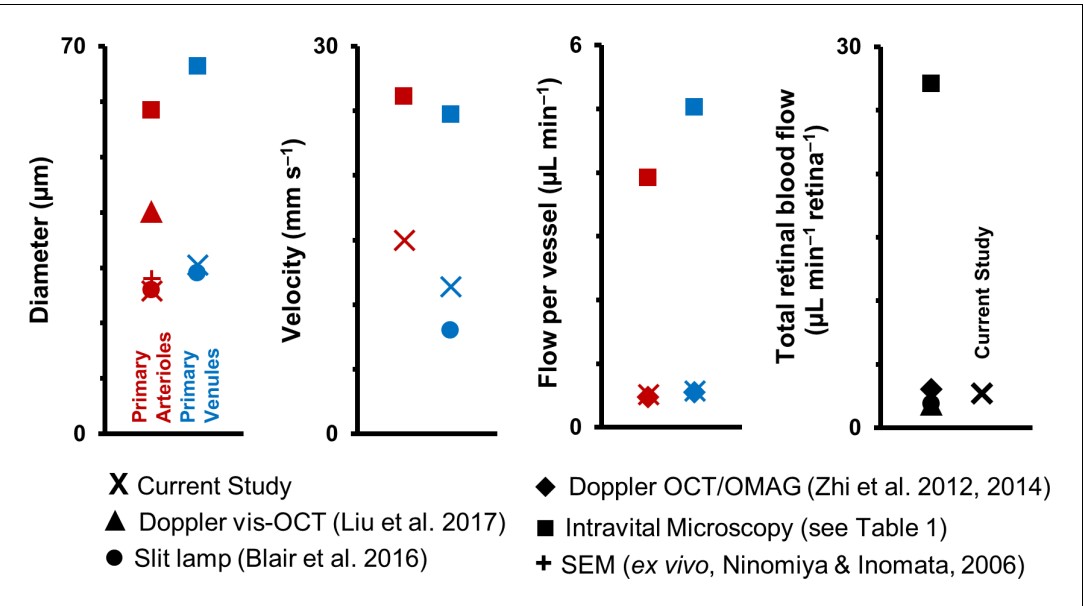

**Figure 11.** Graphical representation of data in *Table 1*. Previous studies of retinal blood vessel diameter, mean velocity and mean flow in first-generation (or 'primary') vessels radiating out of optic disk in normal mice. Wide disparity is observed across studies. (Note: Values for Intravital Microscopy are averaged from *Wright et al., 2012*; *Yadav and Harris, 2011*; *Wang et al., 2011*; *Wang et al., 2010*; *Wright et al., 2009*; *Lee and Harris, 2008*) and *Wright and Harris, 2008*).
DOI: https://doi.org/10.7554/eLife.45077.019

literature was first reported in 2014 by *Bernabeu et al. (2014)* (in a table in their publication). We do not know exactly why such inconsistency exists. While the mouse strain, age and gender differences across studies could contribute to variability, it is unlikely to be the only cause of the large differences in measured values. Using incorrect image magnification on the mouse retina is one possible source of the disparity in measured diameters. Secondly, for mean velocities in primary vessels, intravital microscopy studies used fluorescent microspheres or labelled RBCs, with 10–30 measurements from sparse fluorescent streaks pooled to average out effects of pulsatile and cross-sectional profile variability. The sparse and too few measurements have the potential to under or overestimate the true mean velocity, by either being biased toward particular cardiac phases, or if the incompressible microspheres travel in preferred radial locations in the vessel lumen. Additionally, imaging techniques with insufficient axial resolution can misrepresent diameter, velocity, and flow due to contamination of signal from deeper vessels of the stratified retinal vascular layers. In our study, with ~1 μm *in vivo* lateral resolution and ~10 μm axial resolution, sub-micron correction of eye motion blur, and sampling of all cardiac phases with extremely fast (15 kHz) scanning, we were able to report true mean velocities in these microvessels. Our reported mean velocities in primary vessels were averaged from ~39,000 to 274,000 imaged cells per second.

## Error quantification, limitations and remaining challenges of our study

Our technique measures the component of blood velocity parallel to the *en face* imaging plane. Near the optic nerve head, vessels which are not parallel to the imaging plane can cause errors in flow measurement. This concern is mitigated by restricting our imaging to retinal locations > 170 μm (>5˚ field of view) from the center of the optic disk. First to n[th] generation vessels are largely in plane at these locations. For any remaining non-parallelism at these retinal eccentricities, we provide the following error quantification. For a diving vessel which forms angle γ with the focal plane, there will be an underestimation of velocity, by a factor of $|1 - \cos(\gamma)|$. This error is small for small values of angle γ. For example, for γ = 5˚, measured velocity will be underestimated by 0.38% of the actual value. For reference, the error is <1% for angles (γ) up to 8.1˚. Because the architecture of retinal vessels is predominantly planar (*Paques et al., 2003*), we believe that our technique is well suited to study the complete retinal network, with a potentially lower measurement error.

A limitation of our study is the requirement to sequentially track flow in each vessel. While the maximum field of view of our system of 5˚ enables tracking flow in up to two first-generation vessels simultaneously, and up to five capillaries on the other end of the size spectrum, we tracked flow on a vessel by vessel basis in this study to maximize the pixel density and measurement SNR on each vessel. A remaining challenge of our study is the current inability to track velocity in all first-generation vessels around the optic disk, because of the requirement of an optimal (~5–45˚) angle between the vessel and the 1D scanning beam. While a partial solution for this was implemented by having an additional rotational degree of freedom on the mouse stage, which enable rotating the mouse head and thus the view of the retina by ±30˚, vessels which were near orthogonal to the scanning beam still remained challenging to image. Future work will look at optically rotating the beam so that all vessels around the optic disk can be measured to give a more precise value of total blood flow. We refrained from using the theoretical Poiseuille flow relationships (flow $\propto$ diameter[3] or flow $\propto$ diameter[4]) to predict flow based on lumen diameter of the first-generation vessels in which velocity could not be measured. Our data shows that for the most populous vessels of the mammalian eye, which are tens of microns wide or less, diameter alone is not an accurate predictor of flow (*Figure 10B*).

Finally, we have demonstrated that eye motion minimally impacts our blood velocity measurements in the anesthetized mouse. However, rapid and saccadic eye motion in both the awake behaving mouse and the human being will pose a challenge to our line-scan technique. Such a challenge can be mitigated by restricting the analysis window to epics of relatively low eye motion (gating out saccades/blinks). Only a few seconds of such line-scan data is needed from each vessel to get a reliable measurement of an array of hemodynamic biomarkers. Even 2–4 s of contiguous data will cover multiple cardiac cycles in the human, and our automated algorithm will be able to report velocities from hundreds of thousands of cells.

## Conclusions and future directions

This study has shown that we can noninvasively study an array of functional biomarkers in mouse retinal blood vessels of size 3–46 µm, without using exogenous fluorophores. These microscopic vessels impart a large fraction of the total resistance to blood flow in the body. Therefore, they are often implicated in many a vascular disease, but their small size, tissue scattering properties and aberrations have made them difficult to study *in vivo*.

It has long been proposed that the eye may serve as a window to the brain. The brain is optically inaccessible with high resolution, especially for studying flow in microscopic vessels. Functional imaging of the brain can today only be done with a resolution of a few millimeters, like in fMRI and other techniques, by reporting bulk changes in blood flow as an indirect measure of neuronal activity. The transparency of the eye and it being a part of the CNS makes it a good location to study the behavior of blood flow in a neuronal tissue with high metabolic demand. The optical opacity of the brain and scattering in its tissue requires investigation of its microvasculature to need thinned-skull preparations and fluorescent labeling of either plasma or RBCs for measurement of both velocity and lumen diameter (*Drew et al., 2010*; *Drew et al., 2011*; *Kim et al., 2012*; *Kleinfeld et al., 1998*; *Nishimura et al., 2007*; *Schaffer et al., 2006*; *Shih et al., 2012*). It is also worth mentioning important studies in the rat retina which have achieved measurement of blood flow in a large spectrum of vessel sizes/orders, using fluorescently labeled red blood cells (*Kornfield and Newman, 2014*; *Kornfield and Newman, 2015*). While all the above techniques have made important advances in our understanding of microcirculation in neural tissues, the use of exogenous blood contrast agents raises the concern of maintenance of natural blood perfusion, especially at the capillary level. Furthermore, such invasive techniques cannot be applied to humans. We have shown recently (*Guevara-Torres et al., 2016*) that with label-free adaptive optics imaging, improved contrast of single blood cells can be achieved in the retina, compared to contrast of dye labeling techniques, especially in high flow conditions. Our method of noninvasively studying the retina may serve as a window to supplement the understanding of microcirculation dynamics in the CNS.

The eye may also serve as a window to systemic microvascular health. Vascular diseases like diabetes, atherosclerosis and hypertension are known to impact the microscopic vessels of the body. These tiny vessels often elude *in vivo* study because of their small size, especially in optically inaccessible areas of the body like the heart, kidneys and other internal organs. Our current study of microvessels in the eye opens the door to indirectly studying systemic microvascular disease by peering into the eye. Finally, the noninvasive nature of this study, and the safe (infrared) light levels used, offers the possibility to measure the full spectrum of single blood cell flow in humans (Association for Research in Vision and Ophthalmology [*Joseph et al., 2018*]).

# Materials and methods

**Key resources table**

| Reagent type (species) or resource | Designation | Source or reference | Identifiers | Additional information |
|---|---|---|---|---|
| Strain, strain background (*Mus musculus*) | C57BL/6J mice | The Jackson Laboratory, Bar Harbor, Maine, USA. https://www.jax.org/strain/000664 | RRID:IMSR_JAX:000664, JAX stock #000664 | |
| chemical compound, drug | AK-FLUOR 10% (100 mg mL⁻¹) | Akorn, Lake Forest, Illinois, USA | NDC: 17478-253-10 | IP injection of 0.1 mL of 2.5% weight/volume |

## Animals

Nineteen normal C57BL/6J mice (The Jackson Laboratory stock 000664, Bar Harbor, Maine, USA) with ages ranging from 13 to 73 weeks old were used in this study (11 males, 4 females and four undocumented, weight = 31.7 ± 8.1 g (mean ± SD)). In this study of normal mice, a broad age and sex range were intentionally chosen, to cover the extent of the viable blood flow. Mice were housed in standard cages in a vivarium with 12–hour light/dark cycles. Mice were fed standard chow and water *ad libitum*. All guidelines of University Committee on Animal Resources at the University of

Rochester, Rochester, New York, USA, were followed. Mice were treated according to the Association for Research in Vision and Ophthalmology Statement for the Use of Animals in Ophthalmic and Vision Research.

## Animal preparation for imaging

Mice were anesthetized in two stages, first with an intraperitoneal injection (IP) of ketamine/xylazine (100 mg kg$^{-1}$ ketamine, 10 mg kg$^{-1}$ xylazine), and second with a gas mixture of 0.8–1% (v/v) isoflurane. Supplemental oxygen was delivered through a nose cone to maintain anesthesia throughout the imaging session, which typically spanned ~2 hr. Body temperature was maintained at 37°C using a supplemental heat pad. The pupils were dilated with a drop of 2.5% phenylephrine (Akorn, Lake Forest, Illinois, USA) and 1% tropicamide (Sandoz, Basel, Switzerland). A custom rigid contact lens with +10 D correction and a base curve of 1.6 mm (Advanced Vision Technologies, Lakewood, Colorado, USA) was used in combination with ophthalmic lubricant (GenTeal, Alcon Laboratories, Inc, Fort Worth, Texas, USA) to maintain corneal hydration during *in vivo* imaging. Mice were stabilized on a stereotaxic stage with a bite bar (Bioptigen, Research Triangle Park, North Carolina, USA). *In vivo* imaging could be performed for extended periods of time (typically ~2 hr, with a maximum of ~3 hr). Animals were allowed to recover from anesthesia after imaging and subsequently returned to their cages for use in future experiments. The imaging instrument was in free space, meaning it had no physical contact with the mouse eye or the contact lens. This was critical to maintain normal pressure dynamics of the eye. The three translational degrees of freedom of the stereotaxic stage were used to align the mouse pupil with the exit pupil of the imaging system. Two rotational degrees of freedom were used to navigate to different retinal locations. A third rotational degree of freedom allowed rotating the mouse head along the optical axis, thus enabling rotating the view of the retina at a given retinal location. In a subset of experiments, blood plasma was labelled with an IP injection of 0.1 mL of 2.5% weight/volume of sodium fluorescein (AK-FLUOR 10% (100 mg mL$^{-1}$), Akorn, Lake Forest, Illinois, USA) to confirm lumen diameter of microvessels made with label–free phase-contrast approaches described below.

## AOSLO imaging system

The Rochester mouse AOSLO (adaptive optics scanning light ophthalmoscope) has been described in detail in earlier publications (*Geng et al., 2012*; *Guevara-Torres et al., 2016*). Briefly, the custom-built retinal imaging system has three light sources. A 904 nm laser diode (QFLD-905–10S, QPhotonics, Ann Arbor, Michigan, USA) was used as the wavefront sensing beacon. A 796Δ17 nm super luminescent diode (S790-G-I-15, Superlum, Ireland) was used for near infrared reflectance imaging. A 488 nm laser diode (iChrome MLE, Toptica Photonics, Farmington, New York, USA) was used as the excitation source for sodium fluorescein (collected emission: 520Δ35 nm; FF01-520/35-25, Semrock, Rochester, New York, USA). The light sources were combined into a common light path using appropriate dichroic mirrors and relayed to the eye using reflective optics through a series of five 4-f afocal telescopes. An AOSLO images in *en face* view. At two of the pupil planes in this relay system, a slow 25 Hz galvometric scanner (VM2500+, General Scanning, GSI Group Corp., Massachusetts, USA) and a fast 15 kHz resonant scanner (SC30: 15K, Electro–Optical Products Corp., Glendale, New York, USA) scanned the illumination point spread function (PSF) in orthogonal directions, forming an *en face* rectangular imaging raster on the retinal image plane. The field of view in the two orthogonal directions could be independently changed to vary between 0.5 and 6.2 degrees, and all images were digitized into 640 × 480 pixels with 8–bit grayscale depth. A third pupil plane consisted of a large-stroke high-speed deformable mirror (DM97-15, ALPAO, Montbonnot-Saint-Martin, France) which corrected for low and high order monochromatic aberrations of the eye in real time, measured by a Hartmann-Shack wavefront sensor. The deformable mirror also provided programmable defocus control which enabled focusing the imaging beam on a vascular layer of interest in the retina without using additional optics. The light reflected from the retina followed a similar path as the light into it, and in the detection arm was spectrally split into a near infrared and visible channel to be finally imaged onto photomultiplier tubes (PMTs) (H7422–40 and H7422–50, Hamamatsu, Shizuoka-Ken, Japan) through confocal pinholes (size given below).

### *In vivo* lateral and axial resolution

The confocality of our imaging system provided tight axial sectioning of the retina by rejecting out of focus light, enabling tight focusing on only the retinal vascular layer of interest (superficial, intermediate or deep) (*Schallek et al., 2013a*). The large numerical aperture (NA) of the mouse eye further improved axial sectioning (mouse: NA = 0.49 for 2 mm dilated pupil, human: NA = 0.24 for 8 mm dilated pupil) (*Geng, 2011*). The imaging resolution of this system has been characterized by a previous publication from our lab (*Geng et al., 2012*). Adapting their measured results by linearly scaling for our imaging wavelengths, the measured *in vivo* lateral resolution of our system is 1.2 μm (for 796 nm reflectance imaging) and 0.77 μm (for 520 nm fluorescence emission), giving near diffraction limited imaging *in vivo*. The measured axial resolution is 16.1 μm (for 796 nm reflectance imaging) and 10.5 μm (for 520 nm fluorescence emission). A pinhole of size 2.1 and 4.9 Airy disk diameters in the infrared and visible channels optimized the trade-off between light collected and axial sectioning achieved. The optical point spread function (PSF) size is smaller than the 6.7 μm average size of an undeformed mouse RBC (*Gulliver, 1875*). Combined, these imaging capabilities enabled single blood cell resolution as they flowed in both tight and sparsely packed formations inside microvessels (*Guevara-Torres et al., 2016*). All flowing cells were imaged noninvasively with near infrared backscatter, without the need for foreign dye injections. Furthermore, this detailed resolution enabled sub-micrometer precision when measuring the lumen diameters of retinal vessels.

### Types of blood vessels studied

In total, 123 vessels in 19 mice were studied. Of these, 18 were primary vessels emerging from optic disk (1st generation vessels), 15 were vessels spanning generations of 2nd to 5th branch order in the retina and 90 were single file flow capillaries. A lumen diameter less than 7 μm was used to categorize a vessel as a capillary, as vessels with lumen diameter below the 6.7 μm undeformed size of a mouse RBC (*Gulliver, 1875*) are expected to have single-file flow and facilitate cell deformation that is characteristic of capillary exchange vessels. Using this criterion, lumen diameters of capillaries ranged from 3.2 to 6.5 μm. Of the vessels ≥ 7 μm, 25 were arterioles and eight were venules. Arterioles and venules were identified by inspecting their space-time images to determine direction of flow to or away from the optic disk, as described in detail later. Their lumen diameters ranged from 8.6 to 45.8 μm. Taken together, the range of vessel diameters spanned 3.2–45.8 μm, comprising the smallest to largest vessels in the mouse retinal circulation. Capillaries studied belonged to one of three retinal vascular layers. Arterioles and venules studied were predominantly restricted to the superficial vascular layer. In arterioles and venules, lumen diameter, pulsatile velocity and flow were measured using oblique angle scanning (described below). In capillaries, average cell velocity and average flow rate was measured using orthogonal scanning (*Figure 1—figure supplement 1*, and described later), supplementing the lumen diameter and cell flux reported in single-file-flow capillaries in our previous publication (*Guevara-Torres et al., 2016*).

For the mouse, we define 'large' vessels as first generation vessels near the optic disk, that is vessels that have not yet branched in the retinal plane (diameters: 45.8 to 20.1 μm). We define 'medium-sized' vessels as vessels from second generation and higher, which do not contain single-file-flow (diameters: 20.1 to 7 μm). Small vessels are defined as vessels which have single-file-flow, that is capillaries (diameters: 7 to 3 μm) (*Guevara-Torres et al., 2016*). The average undeformed size of a mouse RBC (6.7 μm) is used as a bio-inspired threshold to identify single-file-flow vessels from others. Conceptually, the above definitions also hold true for the human retina, though the largest human retinal vessels are about four times larger than the largest mouse retinal vessels, to accommodate perfusion for a much greater retinal area. By these definitions, seminal laser Doppler and Doppler OCT studies have measured mean velocity down to 'medium-sized' vessels in the human retina (*Riva et al., 1985*; *Wang et al., 2007*).

### 2D raster imaging for vessel geometry

AOSLO retinal imaging was performed using near infrared illumination (796Δ17 nm, 200–500 μW at cornea) and detection of backscattered/reflected light. The two-dimensional (2D) *en face* imaging spanned 0.5–5.9° field of view, depending on vessel size. The plane of focus was adjusted to axially bisect the widest portion of the vessel, thus ensuring maximum luminal width was investigated based on cylindrical shape of perfused vessels. Most retinal vessels are perpendicular to the imaging

optical axis, making them suitable to study using our *en face* imaging system. The largest retinal vessels, however, dive into the optic disk. Therefore, when imaging primary arterioles and venules emerging from the optic disk, imaging location was between ~170 and 300 μm from the center of the optic disk. A 4 to 20 s video (25 Hz, *Video 1*) was recorded. The average 2D image from the same video was used for analysis of vessel geometry w.r.t. fast scan (angle φ in *Figure 1B* and *Equation 1*), and to create a map of the vascular tree. Correction of eye motion and creation of averaged 2D images are described in subsequent sections.

## Fast 1D line-scan imaging with oblique scanning

To enable high bandwidth velocity measurement, the scan pattern was changed to increase the temporal resolution (*Kleinfeld et al., 1998*; *Zhong et al., 2008*). Briefly, the slow galvometric scanner was stopped at a position such that only the fast one-dimensional (1D) scan imaged the vessel at 15 kHz (*Figure 1*). This 1D scan was placed oblique to the vessel of interest, to determine single-cell velocity in the smallest to largest arterioles and venules in the mouse retina. The mouse head was rotated such that the 1D line passed beyond edge to edge of the obliquely oriented vessel, and such that the angle of intersection between 1D scan and vessel axis was between 6 and 43° (angle φ in *Figure 1B* and *Equation 1*). This angle was positioned manually by the experimental operator and was quantitatively confirmed in post processing using a custom angle determination algorithm based on the Radon transform (strategy discussed in detail below). The plane of focus was kept at the same focus as the 2D scan. The 1D field of view was positioned to traverse the entire luminal diameter whereby least two-thirds of the imaging field covered the vessel (*Figure 1B*). The 1D line ranged from 1.2 to 6.2 ° of visual angle (corresponding to 40.8–210.8 μm on the retina). To determine conversion of pixels to physical retinal coordinates, we used a Ronchi ruling with known grating period and a conversion of 34 μm per degree field of view in the mouse retina (*Schmucker and Schaeffel, 2004*).

## De-warping, correction of eye motion and background removal in retinal images

Both 1D and 2D data were non-linearly distorted in the fast scan direction due to the sinusoidal nature of the 15 kHz resonant scanner. Video frames were de-warped (or de-sinusoided) in real time using previously published technique (*Yang et al., 2015*). 2D retinal motion was corrected in postprocessing using image cross-correlation. Such an approach has been described previously (*Dubra and Harvey, 2010*). The registered video stack was averaged, giving a high signal-to-noise 2D raster image (*Figure 1B*). Only motion in the direction of the fast scanner could be corrected in 1D images as there was no structural information in the orthogonal dimension. In each line-scan sequence lasting 1–10 s, the space-time image was parsed into single time strips ~ 40 ms each. Eye motion in the anesthetized mouse has been previously confirmed to be small in this time regime, relative to blood cell motion (*Guevara-Torres et al., 2016*). A reference strip was manually selected based on vessel centration and lack of motion. For continuous eye motion detection, normalized cross correlation was performed between a reference strip and overalpping target strips, each 40 ms wide, separated by one pixel each (66 μs). The positional shift of each continuous 40 ms time epoch provided a registration signal in the fast scan axis (*Figure 2*). Static features such as the vascular wall are manifest as 'horizontal' lines in space-time images which represent zero velocity. To remove this signal which could confound blood cell analysis, an intensity moving average of the space-time image was computed in the time dimension with a large time window of ~10 ms. This strategy is a high-pass temporal filter, where velocities slower than 0.03 mm s$^{-1}$ are removed via running background subtraction.

## Velocity measurement from 1D line-scan data (oblique scanning)

1D scanning data was recorded for 1–10 s per vessel to capture multiple cardiac cycles. Repeat 1D scans were stacked sequentially to produce a space-time image (*Figure 1D*). RBCs appeared as bright diagonal streaks in such an image as they show their progressive position across the oblique imaging line. The slope of a streak therefore represents an objective measurement of the absolute velocity of each cell according to the equation:

$$\text{velocity}_{\text{cell}} = \frac{\Delta x}{\Delta t} * \sec \phi = k * \cot \theta * \sec \phi \qquad (1)$$

where, $\text{velocity}_{\text{cell}}$ is the local single-cell velocity (mm s$^{-1}$) along the direction of vessel flow, $\Delta x$ (mm) and $\Delta t$ (s) are the respective spatial and temporal displacements of blood cells, whose ratio is given by the slope of the cell streak, $\phi$ is the angle of intersection between flow direction and 1D scan, k is equal to the fast scanner frequency (Hz) multiplied by the number of millimeters of retinal tissue per pixel, and $\theta$ is angle of the cell streak with respect to the space axis in the space-time image. The automated determination of the slope of cell streaks (cot $\theta$) is further described in the following section. In our orientation convention; the more 'vertical' the streak, the faster the cell velocity. The quadrant of $\theta$ (0–90° or 90–180°) gave the direction of flow to or away from the optic disk, enabling categorization of each vessel as arteriole or venule. A 1D scan rate of 15 kHz was used to have a high velocity measurement bandwidth, covering all biologically possible velocities in the largest to smallest vessels in the mammalian retina. The velocity measurement bandwidth was calculated using the relationship:

$$\text{velocity}_{\text{max, measurable}} = \frac{1}{2} * \frac{\delta x}{\delta t} * \sec \phi \qquad (2)$$

where $\delta x$ = sampling density in space, $\delta t$ = sampling density in time (or pixel dwell time) and $\phi$ is the angle of intersection between flow direction and 1D scan (*Drew et al., 2010*). For a typical case with a ~ 5° field of view: $\delta x \approx 0.28$ μm pixel$^{-1}$, $\delta t \approx 0.11$ μs pixel$^{-1}$. The velocity bandwidth was therefore 0.03–1275 mm s$^{-1}$.

## Focal analysis windows (ROIs)

To automatically quantify single-cell velocity, the 1–10 s long space-time image of each vessel was divided into small overlapping regions of interest (ROIs). A single velocity was assigned to each ROI. *Figure 3A* shows a space-time image with a sample ROI size (orange box). For the 33 vessels analyzed with oblique scanning (lumen diameters: 8.6–45.8 μm), the field of view of their space-time images ranged from 1.2 to 6.2° in the space dimension (i.e. 40.8–210.8 μm). The ROI size in these images ranged from 9 to 15 μm in the space dimension. The ROIs were square in pixel space (25–141 pixels) to facilitate a rotationally symmetric Radon sinugram. ROI size was selected to encompass the 'longest' RBC streak seen in space-time images (undeformed size of a mouse RBC of 6.7 μm [*Gulliver, 1875*]). ROIs were advanced with 75% overlap to capture a continuous measurement of local velocity (*Figure 3A*, green boxes). The ROIs were centered every 2–4 μm across the space dimension corresponding to a measurement every 0.25–1.6 μm perpendicular to the vessel lumen when factoring the oblique angle of beam intersection (secϕ factor in *Equation 1*). In the time dimension, ROI size ranged from 1.6 to 9.4 ms, with a measurement provided every 0.4–2.3 ms.

## Automated measurement of blood cell velocity using Radon transform

To quantify cellular-level velocity, the slope of cell streaks in each ROI was determined automatically using a custom algorithm using the Radon transform (MATLAB R2017a, Version 9.2, (with Image Processing Toolbox), MathWorks, Massachusetts, USA; source code provided: *Joseph (2019)* https://github.com/abyjoseph1991/single_cell_blood_flow c3b5f94 (copy archived at https://github.com/elifesciences-publications/single_cell_blood_flow). The Radon transform rotates the input image ROI through a given set of angles and computes the 1D intensity projection of the image for each rotation (*Deans, 2007*; *Drew et al., 2010*; *Bedggood and Metha, 2014*; *Deneux et al., 2011*; *Deneux et al., 2012*; *Chhatbar and Kara, 2013*). The set of 1D projections is compiled as a 2D sinugram image, where each column in the image is a 1D projection at a given angle (*Figure 3B&C*). Each square ROI was cropped circularly before computing its Radon transform, to give equal weight to each possible cell orientation angle. To solve the angle that describes cell slope ($\Delta x/\Delta t$), the standard deviation of pixel values in each 1D intensity projection was computed. The projection angle showing maximum intensity standard deviation corresponded the dominant orientation describing cell velocity in *Equation 1* (*Figure 3D*).

The rotational increment for the Radon transform was chosen to sample a biological range of blood cell velocities with high precision. A linear velocity search space of 0 to 100 mm s$^{-1}$, with a

velocity resolution of 0.1 mm s$^{-1}$ was used by mapping a trigonometrically scaled angle search space for the Radon transform, using *Equation 1*.

## Determination of strength of velocity signal in each analysis window

For each ROI (analysis window), a signal-to-noise ratio (SNR) was determined to avoid reporting velocities in noise–filled analysis windows. The SNR value was defined as the ratio of the maximum column standard deviation divided by the mean column standard deviation of the sinogram image (*Figure 3D*, *Equation 3*).

$$\mathrm{SNR} = \frac{\mathrm{SD}_{\theta_{\max}}}{\sum\limits_{\theta=0°}^{180°} \mathrm{SD}_{\theta}} \tag{3}$$

where, SNR is the signal to noise metric of each ROI, $\mathrm{SD}_{\theta}$ is the standard deviation of the Radon transform at a given search angle $\theta$ and $\mathrm{SD}_{\theta_{\max}}$ is the maximum standard deviation observed across all search angles. We empirically chose a strict threshold of 2.5 to determine whether a streak angle measurement was reliable, based on visual inspection of training data sets. This value is similar to those reported in other studies which use a similar strategy (Drew et al., 2009). Analysis windows that contained SNR <2.5 were ruled as containing insufficient signal and were not included in subsequent analysis.

## Augmented logic gates to prevent spurious measurements

While the SNR alone provides strong rejection of noise-related data, we applied secondary logic gates to further remove false-positives in the space and time dimensions. In the time dimension, a local window spanning twice the length of an ROI in time was measured for mean and variance of each cell velocity reported in that window (n = 9 samples). Velocities which deviated beyond one standard deviation of the mean velocity were ignored. This logic gate is valid as such intervals in time are too small to include significant fluctuations in time that are physiologically relevant (3.2–18.7 ms, or less than 6% of a cardiac cycle). In the space dimension, a logic gate limited velocity reporting to only the position across the lumen that provided a sufficient number of measurements relative to the total– positions that met the SNR criteria for fewer than 2% of the total valid SNR hits were rejected. Combined, the three criteria including SNR (*Equation 3*) and the space and time logic gates focused the analysis to between 3% and 46% of the total searched ROIs. While this percentage may appear low, it still represents thousands of independent measurements in a single vessel. This range indicates a strict and conservative reporting strategy that limits observations to only the conditions of strongest signal. *Figure 4B&C* and *Videos 2* and *3* provide a visual comparison of raw space-time data and measured cell orientations after applying the SNR criterion and the first logic gate. As justified by the mathematical basis above, the visual representation matches subjective determination of blood flow locations in a vessel.

## Graphical visualization of analyzed data

To augment visual feedback to the user, space-time images were superimposed with lines oriented at determined cell angles (*Figure 3E*). The length of these lines represents the ROI size. The color indicates one of two metrics: SNR or velocity. In the case of SNR, line color indicates ROIs that passed (magenta) or failed the SNR criterion (cyan) (*Figure 3E*). When reporting velocity, color indicates jet colormap encoding velocity (*Figure 4C–E*). To report mean velocity and flow rate (33 arterioles and venules in *Figure 10A&B*), 1 s of space-time image data was analyzed and velocities from 499 to 25210 ROIs were reported. To report the average cardiac cycle and velocity profile as a function of cardiac phase for the arteriole in *Figures 4* and *10* s of data was analyzed, and velocities from 123,761 ROIs reported.

## Measurement of pulsatile flow and vascular indices

All measures performed at the same point in time were averaged to reveal attributes such as cardiac cycle. The time binning window was chosen to span 15 ms, to sample each cardiac cycle 10–27 times for an expected heart rate range of 150–400 beats per minute, thus sampling well above the Nyquist limit. The binned velocities were plotted as a function of time (*Figure 4F*). After peaks and troughs

of each cardiac cycle were identified, all cardiac cycles were phase-matched and averaged to produce a high signal-to-noise average cardiac cycle (*Figure 4G*). The heart rate was measured reporting the inverse of the average cardiac cycle time period. The maximum, minimum and mean velocities of the average cardiac cycle ($v_{max}$, $v_{min}$ and $v_{mean}$ respectively) were used to compute vascular indices of pulsatility and resistivity according to the formulae:

$$\text{Pulsatility index (PI)} = \frac{v_{max} - v_{min}}{v_{mean}} \tag{4}$$

$$\text{Resistivity index (RI)} = \frac{v_{max} - v_{min}}{v_{max}} \tag{5}$$

The time positions of the maximum and two minimum velocities ($t_{max}$, $t_{min1}$ and $t_{min2}$ respectively, orange and green points in *Figure 4G*) were used to compute a custom metric of asymmetry in the profile of the average cardiac cycle:

$$\text{Asymmetry index (AI)} = \frac{t_{min2} - t_{max}}{t_{max} - t_{min1}} \tag{6}$$

## Measurement of cross-sectional velocity profile

Measured velocities in a single vessel were reported as a function of position along the vessel lumen (velocity profile) by averaging, across time, all measures performed at a specific radial location. The spatial binning window was chosen to be 4 µm, which corresponds to intervals of 0.77 µm across the vessel lumen when factoring vessel angle (*Figure 4H* and *Figure 6*). Additionally, instantaneous velocity profiles were phase locked to the cardiac cycle to reveal dependence of the profile on cardiac phase (*Figure 4H* and *Video 3*).

The measured profiles were fit with several models of blood flow (*Figure 6B*). Symmetric flow was assumed in all models by constraining the fit to the position of maximum measured velocity, and an equal number of data points to each side of it. The RBC flow profile was modelled as described by *Zhong et al. (2011)*.

$$V(r) = V_{max}\left[1 - (1 - \beta).\left|\frac{r}{R}\right|^B\right] \tag{7}$$

where, $V(r)$ is velocity at radial position r in a vessel, R is the maximum radial position (half of measured lumen diameter), $V_{max}$ is the maximum velocity (or centerline velocity, assuming symmetric flow), $\beta$ is a custom factor that is proportional to the normalized extrapolated RBC velocity at the lumen edges, and B is a custom bluntness index. Additionally, the plasma velocity profile was modeled assuming zero velocity at the lumen edges, to satisfy the no-slip boundary condition (*Figure 6B*).

## Orthogonal 1D scan to measure velocity in single-file flow capillaries

In a previous publication, we reported label-free measurement of single-cell flux (cells s$^{-1}$) in capillaries with single-file flow (*Guevara-Torres et al., 2016*) (*Figure 1—figure supplement 1*). To compare those measurements with flow rate (µL min$^{-1}$) measurements of larger vessels in this current study, we measured the average cell velocity from 1 s long space-time images of 90 capillaries captured with split detection phase contrast (*Figure 10C*). To determine cell velocity, cell locations were marked and were phase registered to produce an 'average cell'. The temporal width of this average cell was used to report average capillary velocity using the equation reported by us previously (*Guevara-Torres et al., 2016*).

$$\text{velocity}_{mean} = \frac{d}{t} \tag{8}$$

where $\Delta t$ is the temporal width of the average cell, and d is the average length of RBCs along the direction of flow. RBCs deform during single-file flow, with a typical long-to-short axis ratio of 2:1. Considering this deformation and a mean plasma layer of ~0.5 µm, mean RBC lengths ranging from 4.5 to 10 µm were reported in a capillary glass tube study (*Hochmuth et al., 1970*) across a variety

of capillary inner diameters. A value of 7.25 μm was assumed for 'd' in this publication, for all capillaries studied (lumen diameter range: 3.2–6.5 μm).

## Lumen diameter measurement

Lumen diameter was measured using 796 nm backscatter in the split-detection modality (*Figure 7A*), (*Chui et al., 2014*; *Chui et al., 2012*; *Guevara-Torres et al., 2016*; *Sulai et al., 2014*). The 2D raster videos were registered for eye motion and then used to compute the temporal standard deviation or motion contrast image (*Chui et al., 2014*; *Sulai et al., 2014*) (*Figure 7B*). To confirm the label free measurement represented the true lumen diameter of the arterioles and venules, diameter was also measured simultaneously using conventional fluorescein contrast (*Figure 7C*,~50 μW of 488 nm excitation, 520Δ35 nm emission). For diameter measurement, the vessel axis orientation was determined using the Radon transform. Multiple 5 μm strips orthogonal to the length of the vessel were analyzed for diameter (*Figure 7D–F*). For arterioles and venules, the average diameter was reported from 2 to 31 five-micron strips along the length of the vessel. Lumen diameter measured distance between vascular walls in split detection images, corresponding to half-height points in the split-detection intensity image (*Figure 7G*). Independently, the standard deviation (SD) image that computes pixel variance over time revealed the erythrocyte column by way of motion contrast. Lumen edges were measured as the inner boundary of the motion contrast image (*Figure 7H*). For lumen measures using sodium fluorescein, lumen edges were identified as the half-height points, using the peak intensity of the profile as reference (*Figure 7I*). Lumen diameters for all 90 capillaries were measured in our previous publication (*Guevara-Torres et al., 2016*) using split-detection contrast to measure inner wall to inner wall distance.

## Calculation of mean flow rate per vessel

The mean RBC velocity (over 1 s of space-time data) and lumen diameter for each arteriole and venule were combined to give its mean flow rate (in μL min$^{-1}$) according to Poiseuille's equation (*Figure 10B*):

$$\text{Blood flow rate} = \text{Velocity}_{\text{mean}} \times \frac{\pi \times \text{Diameter}^2}{4} \qquad (9)$$

Flow rate in capillaries was also measured their average cell velocity and cross-sectional area in the above equation (*Figure 10D*). Our previous publication had reported flux (cell s$^{-1}$) in these capillaries (*Figure 10E*) (*Guevara-Torres et al., 2016*).

## Statistics

Data are reported as mean (standard deviation) unless otherwise mentioned. Shaded regions in plots represent the mean ±1 SD, unless otherwise mentioned. The number of samples used in variation is given wherever possible. All analyses were performed using MATLAB R2017a (The MathWorks, Inc, Massachusetts, USA).

# Acknowledgements

We are grateful to Qiang Yang, Jie Zhang, Chas Pfeifer and Jennifer Strazzeri for their technical contributions to this work. We are very grateful to Robin Sharma for critically reviewing the original manuscript and providing insightful suggestions. We thank Kosha Dholakia, Colin Chu, Josie Lorenzo, Emmanuel Alabi and Guanping Feng for their feedback on the manuscript. We thank Keith Parkins for providing images of simulated blood cells for the eLife Digest cover art.

# Additional information

### Competing interests

Aby Joseph, Andres Guevara-Torres: Was supported by a research grant from Hoffman-La Roche Inc. Has also submitted patent applications (held through University of Rochester) in the provisional stage on the approach of imaging and measuring blood cell speed automatically and in real-time

using adaptive optics. Jesse Schallek: Was supported by a research grant from Hoffman-La Roche Inc. Holds a patent on the AOSLO blood cell imaging technology held by the University of Rochester: Patent #9,844,320 Issued: 12/19/2017, "System and Method for Observing an Object in a Blood Vessel". Has also submitted patent applications (held through University of Rochester) in the provisional stage on the approach of imaging and measuring blood cell speed automatically and in real-time using adaptive optics.

## Funding

| Funder | Grant reference number | Author |
| --- | --- | --- |
| National Eye Institute | EY028293 | Aby Joseph<br>Andres Guevara-Torres<br>Jesse Schallek |
| Research to Prevent Blindness | Career Development Award | Aby Joseph<br>Andres Guevara-Torres<br>Jesse Schallek |
| Dana Foundation | David Mahoney Neuroimaging Award | Aby Joseph<br>Andres Guevara-Torres<br>Jesse Schallek |
| Roche | pRED | Aby Joseph<br>Andres Guevara-Torres<br>Jesse Schallek |
| National Eye Institute | P30 EY001319 | Jesse Schallek |
| Research to Prevent Blindness | Unrestricted Grant to the University of Rochester Department of Ophthalmology | Jesse Schallek |
| Research to Prevent Blindness | Stein Innovation Award | Aby Joseph<br>Andres Guevara-Torres |

The funders had no role in study design, data collection and interpretation, or the decision to submit the work for publication.

## Author contributions

Aby Joseph, Conceptualization, Data curation, Software, Formal analysis, Validation, Investigation, Visualization, Methodology, Writing—original draft, Writing—review and editing; Andres Guevara-Torres, Data curation, Formal analysis, Methodology; Jesse Schallek, Conceptualization, Resources, Supervision, Funding acquisition, Validation, Investigation, Methodology, Project administration, Writing—review and editing

## Author ORCIDs

Aby Joseph https://orcid.org/0000-0001-8143-801X
Jesse Schallek http://orcid.org/0000-0002-6337-4187

## Ethics

Animal experimentation: All guidelines of University Committee on Animal Resources at the University of Rochester, Rochester, New York, USA, were followed. PHS Assurance #D16-00188(A3292-01). Reference number from University Committee on Animal Resources is 101017 and protocol number is 2010-052. Mice were treated according to the Association for Research in Vision and Ophthalmology Statement for the Use of Animals in Ophthalmic and Vision Research.

## Decision letter and Author response

Decision letter https://doi.org/10.7554/eLife.45077.027
Author response https://doi.org/10.7554/eLife.45077.028

# Additional files

### Supplementary files
• Supplementary file 1. The number and type of mice used in each figure. Our study is focused on (1) the method and (2) generating a novel dataset of hemodynamics in the healthy mouse. As such, *Figures 1–9* are designed to show the method and analysis steps, algorithm validation and representative examples of measurements made. Therefore, these figures typically show one or a few vessels, from one mouse, unless otherwise mentioned in the table above. The population data across all mice imaged is shown in *Figure 10*. Population data are reported for mean velocity, flow, diameter and flux, across multiple mice, as quantified in the table above.
DOI: https://doi.org/10.7554/eLife.45077.020

• Supplementary file 2. Raw space-time image corresponding to top-half of *Video 2*. ~1 s of high-resolution data of single-cell blood flow captured in the 25.3 µm arteriole shown in *Figure 4*. Scaling given in *Video 2* legend.
DOI: https://doi.org/10.7554/eLife.45077.021

• Supplementary file 3. Cell slopes and velocity overlaid on the original space-time image in *Supplementary file 2*. N ≈ three unique cardiac cycles shown.
DOI: https://doi.org/10.7554/eLife.45077.022

• Transparent reporting form
DOI: https://doi.org/10.7554/eLife.45077.023

### Data availability
The raw AOSLO data is large in size, constituting 100s of GBs of data. One representative file is provided so that users can see raw data format and resolution (see video 2) and a single subject representative data set has been made available via Zenodo (https://doi.org/10.5281/zenodo.2658767). The full data set can be provided on request to the corresponding author.

The following dataset was generated:

| Author(s) | Year | Dataset title | Dataset URL | Database and Identifier |
|---|---|---|---|---|
| Aby Joseph, Andres Guevara-Torres, Jesse Schallek | 2019 | AOSLO Single Cell Blood Flow - Raw Data (eLife paper: Joseph et al. 2019) | https://doi.org/10.5281/zenodo.2658767 | Zenodo, 10.5281/zenodo.2658767 |

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
