## [Decision Letter]

Thank you for submitting your manuscript "Imaging single-cell blood flow in the smallest to largest vessels in the living retina" to *eLife*. Your article has been reviewed by three peer reviewers, and the evaluation has been overseen by a Reviewing Editor and a Senior Editor. As you will see, all of the reviewers were impressed with the importance and novelty of your work.

The three reviews are included in this letter as there are a variety of specific and useful suggestions in them. In particular, reviewer #2 raises important questions about the validation of the automated algorithm. We appreciate that the reviewers' comments cover a range of suggestions for improving the manuscript.

*Reviewer #1:*

The authors have used AOSLO imaging with ultrafast camera capture to image blood cells in the full range of blood vessels in the eyes of anesthetized mice. Automated analysis enabled measurement of thousands of blood cells per second. The study is important because it provides the measurements in a normal mouse for future comparison with measurements in mice with retinal diseases. I do not have the expertise to validate all the methods, equations, and calculations used to generate the data- hopefully you have selected other reviewers equipped to do so. Assuming that validation occurs, the potential value of the normative database and the comparison with future measurements in diseased eyes is large and worthy of publication.

*Reviewer #2:*

The authors present an impressive system for measuring hemodynamics across a range of blood vessel sizes in the living mouse eye. This study combines approaches from technical instrumentation, computer algorithm development with application to mouse and vascular biology. This type of approach is likely to be influential in the field of retinal imaging.

The AOSLO system seems to have been previously developed and there is limited improvement on the technical instrument. The novelty of the work stems from development of automated computer algorithms and in the novel dataset of mouse hemodynamics that was generated. To this end the primary concern is that the automated algorithm was not fully or properly validated, so it is difficult for the reader to assess how accurate the algorithm truly is. There should be some comparison to ground truth or other systematic careful validation to show that the automated results are sound. An artificial dataset could also be used in which the authors simulate velocities and velocity changes from empirical data and establish that their algorithm can properly detect these changes.

The overall organization of the article be improved so that the most relevant results are presented first followed by Discussion and Materials and methods at the end of the paper. Overall, the results of the paper are impressive.

The overall logic flow of the paper could be improved particularly with regards to specifying, motivating and justifying the number and type of mice used for each subexperiment within the overall paper. Currently, the study design is difficult to follow as it seems to be a blend of cross sectional and longitudinal imaging (or if not it is very ambiguous). This should be clarified through the use of supplementary tables. Given that this data was collected over a period of 3 years (2014-2017), there needs to be supplementary tables that clearly lay out the number and type of mouse used for each analysis as it is very difficult to assess how many mice were used in each section of the paper.

In the introduction, the authors should more clearly motivate why an in vivo mouse system needs to be developed as it seems that similar techniques could or have already been developed for the human eye, so that it seems backwards to go from human back to mouse.

Generally, a description of limitations should be more clearly emphasized. A key item missing is a quantitative assessment of error and sources of error including what, if any, the effect of vessels not staying in the same focal plane might have. Near the optic nerve head, the vessels dive down through the retinal layers and this could have an effect on speed measurements. Carefully quantifying the expected error and propagating it through the measurements would be helpful as part of the validation process that is missing.

*Reviewer #3:*

The retinal microcirculation provides a unique possibility to observe the microcirculation noninvasively in animals and humans. Such observation is performed through the lens of the eye, which has imperfect optical properties, limiting the resolution that can be achieved. In this work, the authors show how the use of adaptive optics, a technique pioneered in for use in astronomy, can lead to imaging of red blood cell motion in vessels from the primary vessels at the optic disk down to the level of capillaries. This is a significant technical advance, with potentially high impact on the studies of retinal microcirculation for research and clinical applications. Although the emphasis is on the method, the quantitation of mouse retinal hemodynamic parameters presented here provides baseline data that will be useful for future work. The presentation of the work is clear and thorough (if perhaps a little verbose at times).

---

## [Author Response]

Reviewer #1:The authors have used AOSLO imaging with ultrafast camera capture to image blood cells in the full range of blood vessels in the eyes of anesthetized mice. Automated analysis enabled measurement of thousands of blood cells per second. The study is important because it provides the measurements in a normal mouse for future comparison with measurements in mice with retinal diseases. I do not have the expertise to validate all the methods, equations, and calculations used to generate the data- hopefully you have selected other reviewers equipped to do so. Assuming that validation occurs, the potential value of the normative database and the comparison with future measurements in diseased eyes is large and worthy of publication.

Thank you! We have addressed in detail the validation and ground-truth aspects of our algorithm – please see response to reviewer #2.

Reviewer #2:The authors present an impressive system for measuring hemodynamics across a range of blood vessel sizes in the living mouse eye. This study combines approaches from technical instrumentation, computer algorithm development with application to mouse and vascular biology. This type of approach is likely to be influential in the field of retinal imaging.The AOSLO system seems to have been previously developed and there is limited improvement on the technical instrument. The novelty of the work stems from development of automated computer algorithms and in the novel dataset of mouse hemodynamics that was generated. To this end the primary concern is that the automated algorithm was not fully or properly validated, so it is difficult for the reader to assess how accurate the algorithm truly is. There should be some comparison to ground truth or other systematic careful validation to show that the automated results are sound. An artificial dataset could also be used in which the authors simulate velocities and velocity changes from empirical data and establish that their algorithm can properly detect these changes.

We largely agree with reviewer #2 that a complete validation would provide indisputable evidence of the algorithm accuracy. While we address precision, accuracy is a challenging problem that requires a “ground truth” data set. There are substantial challenges with this request, nevertheless, we provide the following response:

1) A true “ground truth” data set showing biological single cell velocity does not exist for the living eye. Moreover, we feel that our approach is a major step forward in accomplishing this goal. No such biological dataset with validated velocities exist to our knowledge.

2) Despite lack of biological ground truth, we provide a mathematically complete set of equations (Equations 1-9) with corresponding instruction of how those equations are used to measure velocity. In this way, a mathematically inclined researcher can directly apply our strategy to test validity of these approaches. In the spirit of *eLife* communication, we provide a complete work flow and also provide fully documented source code for all our calculations. All assumptions and variables are declared in the Materials and methods section.

3) While “ground truth” datasets do not exist, we do go through sufficient length to compare our velocity measures against an exhaustive list of other imaging modalities that report flow and velocity in the mouse. Table 1 of our paper shows a survey of the studies that use intravital microscopy, Doppler OCT and OMAG OCT to determine blood flow in the mouse (Figure 11). In absence of ground truth (which notably was also lacking in all of the aforementioned studies), we find that our solution converges on similar measures of blood velocity and total vessel flow providing an additive reassurance that the approach is sound. Additionally, our *in vivo* blood vessel diameters (necessary to compute flow) are confirmed with a number of previous *in vivo* studies in the mouse. There is one notable discrepancy in diameter reporting in the literature, and thus we confirmed our measure with the histological gold-standard of scanning electron microscopy. We discuss in our manuscript the similarities (and also discrepancies) of our values against a fairly complete survey of relevant previous literature in the mouse retinal circulation.

After consideration of the above, we recognize that arguments 1-3 are somewhat passive to the specific request of ground truth. We have addressed this question of ground truth head-on in a tandem manuscript that is being submitted to Journal of Computational Neuroscience. In this tandem publication, we produce a data phantom that simulates single blood cells moving in a vessel. In this case, the true velocity of single particles is known and can be systematically varied to test algorithm performance at varying speeds. In brief, our Radon algorithm recovers the velocity with high degree of accuracy against the known and masked phantom values (R^2^=0.9991).

We hope that reviewer #2 and the editors will agree that this analysis is best suited for a separate publication as Materials and methods, Results, Discussion and interpretation would add to an already long 40-page manuscript under consideration at *eLife*. In our opinion, inclusion of this detailed analysis clutters our current manuscript and detracts from the central purpose of sharing this novel (and complete) framework with our peers.

The overall organization of the article be improved so that the most relevant results are presented first followed by Discussion and Materials and methods at the end of the paper. Overall, the results of the paper are impressive.

Thank you! We have re-ordered the article to have Results first, followed by Discussion and Materials and methods.

The overall logic flow of the paper could be improved particularly with regards to specifying, motivating and justifying the number and type of mice used for each subexperiment within the overall paper. Currently, the study design is difficult to follow as it seems to be a blend of cross sectional and longitudinal imaging (or if not it is very ambiguous). This should be clarified through the use of supplementary tables. Given that this data was collected over a period of 3 years (2014-2017), there needs to be supplementary tables that clearly lay out the number and type of mouse used for each analysis as it is very difficult to assess how many mice were used in each section of the paper.

Building on the reviewer’s suggestion, we provide clarity on the total subjects and data relevant to each figure in a new table (Supplementary file 1). This table summarizes the number and type of mice used in each sub-experiment/analysis. In the caption of this table, we have motivated and justified the numbers used, as requested.

*In the Introduction, the authors should more clearly motivate why an* in vivo *mouse system needs to be developed as it seems that similar techniques could or have already been developed for the human eye, so that it seems backwards to go from human back to mouse.*

We have now made the motivation clearer:

“In this study, we use the living mouse to benchmark the automation of blood velocity data. […] Development on quantification of mouse blood flow approaches will be directly applicable to clinical approaches using the same instrumentation.”

Generally, a description of limitations should be more clearly emphasized. A key item missing is a quantitative assessment of error and sources of error including what, if any, the effect of vessels not staying in the same focal plane might have. Near the optic nerve head, the vessels dive down through the retinal layers and this could have an effect on speed measurements. Carefully quantifying the expected error and propagating it through the measurements would be helpful as part of the validation process that is missing.

The reviewer brings to light an important point worth discussing – that there is possible error due to unaccounted for axial component of velocity. We include additional descriptions of the errors, assumptions and limitations of the approach, which are consolidated in one section in the revised article (reproduced below). In addition to the text provided, we directly address that the retinal circulation in the mouse, outside of the optic disk, is largely in plane with the en face imaging plane (Paques et al., 2003) (unlike the neocortex, where vessels are highly non-planar). Therefore, the error associated is expected to be small, especially when compared to the physics of how a Doppler signal is generated. For example, laser and ultrasound Doppler measure the axial component of velocity, and the en face plane of a majority of retinal vessels provides a relatively weak Doppler signature.

Upon this helpful suggestion, we quantify the error and discuss in the manuscript as follows:

“Our technique measures the component of blood velocity parallel to the en face imaging plane. […] Because the architecture of retinal vessels is predominantly planar (Paques et al., 2003), we believe that our technique is well suited to study the complete retinal network, with a potentially lower measurement error.”

We contrast this to the error in some Doppler techniques, where the measurement signal arises from the diving/rising component of velocity (which is orthogonal to the velocity component measured in our technique). Therefore, the measurement error is equal to |1 – sin(γ)| for techniques which measure the out-of-plane velocity component. For this reason, many such techniques are most accurate at positions where vessels are not en face, often closer to the optic disk, where the diving/rising angle (γ) is as high as possible.

Additionally, further discussion is provided in the subsection “Velocity measurement algorithm and its key assumptions”.